# SELF-CORRECTION BENCH: UNCOVERING AND ADDRESSING THE SELF-CORRECTION BLIND SPOT IN LARGE LANGUAGE MODELS

## ABSTRACT

Although large language models (LLMs) have transformed AI, they still make errors and follow unproductive reasoning paths. Self-correction is vital for safety-critical applications. However, we uncover a failure: LLMs can correct errors (by fixing external ones) but fail to activate this capability for identical internal errors - a limitation we term the **Self-Correction Blind Spot**. To study this, we introduce **Self-Correction Bench**, an evaluation framework that isolates self-correction behavior from knowledge limitations through controlled error injection. Testing 14 open-source non-reasoning models shows a 64.5% average blind spot rate. We show robustness in mathematical reasoning across complexities, and extend to closed-source models, non-mathematical domains, and on-policy errors. Causal evidence links this to training data: human demonstrations lack error-correction sequences, but fine-tuning with them reduces the blind spot. Appending a simple "*Wait*" prompt cuts blind spots by 89.3%, revealing latent capabilities. Our work exposes a training-induced limitation and provides practical fixes to boost LLM reliability in critical domains.

## 1 INTRODUCTION

Large Language Models (LLMs) have rapidly advanced natural language processing, achieving state-of-the-art results on a diverse range of tasks (OpenAI et al., 2024; Anthropic, 2024a; Gemini Team, 2025; Yang et al., 2025; Meta, 2025; DeepSeek-AI et al., 2025a). However, despite their impressive capabilities, LLMs are known to exhibit unpredictable failures and generate inaccurate information (Maynez et al., 2020; Huang et al., 2025; Bang et al., 2023; Shi et al., 2023), or explore an unproductive reasoning path and commit to it. A particularly concerning issue is that LLMs can make errors even in simple tasks (Nezhurina et al., 2025), despite possessing the necessary underlying knowledge to provide the correct solutions, raising reliability concerns that hinder deployment in critical applications.

Studying LLM self-correction behavior in natural settings is challenging due to their inherent accuracy; the rarity of naturally occurring errors makes systematic diagnostic and evaluation challenging. Our central question: Given that a model can correct an error when presented externally (proving it has the capability), does it activate this capability when the identical error is presented to itself? This isolates activation failures from knowledge gaps. To study this, we construct Self-Correction Bench by systematically injecting errors into both the user prompt (defined as an external error) and the model generation (defined as an internal error). Our off-policy error injection design enables this distinction by keeping the error constant while varying only the attribution.

Our results reveal that LLMs fail to correct internal errors (64.5% average failure rate), but reliably fix identical errors from external sources. We refer to this phenomenon as the Self-Correction Blind Spot. This rules out knowledge deficiency as the root cause; instead, the blind spot stems from a lack of activation for self-correction. Strikingly, appending a simple "*Wait*" serves as a diagnostic probe, reducing the blind spot by 89.3%, confirming a minimum prompt can unlock latent correction abilities.

We provide a systematic understanding of why the Self-Correction Blind Spot exists, supported by analysis of correction marker patterns in post-training data and causal evidence that including error-

correction traces in supervised fine-tuning (SFT) substantially mitigate the blind spot, explaining the superior performance of reasoning models.

We further validate the robustness of this finding in closed-source frontier models, non-mathematical domains (logic, object tracking), and on-policy errors.

Our contributions are threefold.

- Discovery of Self-Correction Blind Spot: a systematic failure of LLMs to correct internal errors despite competency on external ones. We provide causal evidence that this is driven by post-training data biases where human demonstrations rarely include self-correction sequences.

- Self-Correction Bench: a controlled evaluation framework with error-injected reasoning traces that isolates the self-correction behavior from internal knowledge limitations, for cross-model comparison.

- Training recipe solution: causal evidence that including error-correction traces in SFT mitigates the blind spot, with "*Wait*" as a probe demonstrating activation is the limiting factor.

These results advance both our understanding of LLM reasoning flaws and provide a practical solution to improve their reliability in real-world use.

## 2 RELATED WORK

**Intrinsic self-correction in LLMs**. Recent work explores intrinsic self-correction via self-feedback (Shinn et al., 2023; Madaan et al., 2023; Kim et al., 2023; Kamoi et al., 2024b) or critic ensemble (Mousavi et al., 2023), but limitations persist. Feedback quality suffers without oracle labels (Huang et al., 2024): prior studies attribute this to poor error localization (Tyen et al., 2024) and detection (Kamoi et al., 2024a). Most approaches use multi-step prompting, whereas we focus on single-pass self-correction and study limitations from a cognitive perspective. Related work using RL for self-correction (Kumar et al., 2025) or training signals from ground truth (DeepSeek-AI et al., 2025a) contrasts with our test-time, no-fine-tuning approach.

**Prompt injection for evaluation**. Traditional prompt injection research focuses on adversarial manipulation (e.g., attackers injecting malicious instructions to distort outputs) (Wei et al., 2023; Liu et al., 2024). Controlled error injection to evaluate self-correction is underexplored. For example, Lanham et al. (2023) injected mistakes into reasoning chains to measure consistency between steps and conclusions, but not self-correction capability. Our work advances this by systematically injecting errors across task complexities to reveal uncharacterized blind spots in how LLMs correct themselves.

**Hallucination snowballing**. Zhang et al. (2024) demonstrate that once LLMs hallucinate, subsequent tokens often align with the initial error, a "snowball" effect, suggesting inherent limits to self-correction during generation. We explain this phenomenon by identifying Self-Correction Blind Spot: LLMs reliably correct errors in external inputs, but fail to correct errors in their own outputs. This distinction is critical to understanding why snowballing persists.

**Test-time interventions**. Recent efforts have shifted compute from training to test time (Snell et al., 2025), yielding improved performance (e.g. Muennighoff et al. (2025) appends "Wait" to force longer reasoning traces, but improvement mechanisms remain understudied. We show interventions activate dormant self-correction capabilities in unfine-tuned models, improving performance on error-prone tasks.

**Cognitive bias in LLM**. LLMs exhibit human-like cognitive biases (Koo et al., 2024; Echterhoff et al., 2024; Jones & Steinhardt, 2022), and we link the bias blind spot (the tendency to overlook one's own biases) (Pronin et al., 2002) to impaired self-correction. This connects high-level cognitive limitations to the fine-grained failure mode we characterize.

Our work integrates these threads into a systematic methodology for testing self-correction, reveals that LLMs suffer from a blind spot (inability to correct internal errors) despite having the knowledge.

## 3 CONCEPTUAL MOTIVATION

Building on these insights, we now formalize the theoretical framework underlying our empirical investigation. We provide conceptual motivation for our empirical study, focusing on error states, self-correction mechanisms, and their measurement.

### 3.1 ERROR AND SELF-CORRECTION: THE CASE FOR MARGINALIZATION

Autoregressive LLMs cannot guarantee every generated token is correct as the number of token grows, resulting in hallucination (Maynez et al., 2020), snowballing errors (Zhang et al., 2024), or unproductive reasoning path or execution flaws. Thus, self-correction is necessary for robustness: models must reverse errors to produce a correct answer. Note that a correct answer does **not** require all previously generated tokens to be correct, as one might be concerned only with the final answer.

To formalize this, let $\mathcal{E} = \{e_0, e_1, \ldots, e_k\}$ denote a set of mutually exclusive and collectively exhaustive discrete error states, where $e_0$ represents the "no error" state, and $e_1, \ldots, e_k$ represent distinct error conditions. For each state $e_i \in \mathcal{E}$, let $R_{e_i}$ denote the response set. The probability of a model, $M$, giving a correct answer can be marginalized over error states:

$$P_M(r_{correct}) = \sum_{e \in \mathcal{E}} P_M(e) \cdot P_M(r_{correct}|e) = \sum_{e \in \mathcal{E}} \sum_{r_m \in R_e} P_M(e) \cdot P_M(r_m|e) \cdot P_M(r_{correct}|r_m, e),$$

(1)

where $r_m$ is the model's response, and $P_M(r_{correct}|r_m, e)$ captures self-correction of $r_m$. Here, $P_M(r_{correct})$ depends critically on self-correction: even with frequent errors, high $P_M(r_{correct}|r_m, e)$ can yield strong performance. Error-free generation is a **special** case of this framework - not the only path to correctness.

### 3.2 EXTERNAL AND INTERNAL SELF-CORRECTION, AND SELF-CORRECTION BLIND SPOT

We distinguish self-correction by error source:

1. **Internal correction**: Metacognitive monitoring of the model's initial response $r_m$.
2. **External correction**: Evaluation of errors in the user prompt $r_u$.

This distinction is motivated by the cognitive bias, "bias blind spot". Pronin et al. (2002) show that humans are able to identify cognitive biases in others while failing to see those same biases in themselves, suggesting LLMs trained on human demonstration might share this limitation.

To quantify this, we define the Self-Correction Blind Spot as:

$$\text{Self-Correction Blind Spot} = \begin{cases} 1 - \frac{P_M(r_{correct}|r_m, e)}{P_M(r_{correct}|r_u, e)} & \text{if } P_M(r_{correct}|r_u, e) > 0 \\ 0 & \text{if } P_M(r_{correct}|r_u, e) = 0 \end{cases}$$

(2)

A value of 1 indicates a total blind spot: the model can correct external errors but not internal ones. By design, Self-Correction Blind Spot **isolates activation failure from confounding factors**. This conditionality is why off-policy error injection is essential: it ensures we measure whether models activate capabilities they provably possess, not whether they have the capabilities at all.

### 3.3 CONTROLLED ERROR INJECTION: MEASURING SELF-CORRECTION IN PRACTICE

The marginalization framework (Equation 1) is intractable in practice: $P_M(e)$, the true probability of error states, is unobservable as LLMs operate over infinite prompt spaces. To solve this, we introduce controlled error $e_{controlled}$. For internal correction, we inject an incorrect partial response into the model's "own" output (omitting stop tokens to allow continuation/self-correction); for external correction, we inject the same error into the user prompt instead. We empirically estimate $P_M(r_{correct}|r_m, e_{controlled})$ and $P_M(r_{correct}|r_u, e_{controlled})$ by measuring $P_M(r_{correct})$ when controlled error is injected in $r_m$ and $r_u$ respectively, which forms the basis of our Self-Correction Bench.

This framework - marginalizing over errors, distinguishing internal/external correction, and using controlled injection - provides the conceptual underpinning for our empirical analysis.

## 4 SELF-CORRECTION BENCH

Having established the theoretical foundation, we introduce a benchmark to measure Self-Correction Blind Spot in LLMs across varying task complexities, enabling diagnosis of where correction fails. By progressing from simple to realistic errors, we isolate confounding factors such as knowledge limits or reasoning complexity. Our benchmark comprises "Self-correct Like I am 5" (SCLI5), GSM8K-SC and PRM800K-SC, with controlled error injection to compare performance under identical conditions. See Table 5 in Appendix for details of dataset.

For each dataset, we systematically inject an identical error into both model response ($r_m$) and user prompt ($r_u$), allowing empirical estimation of both correction probabilities $P_M(r_{correct}|r_m, e_{controlled})$ and $P_M(r_{correct}|r_u, e_{controlled})$ under identical error conditions. Unlike Zheng et al. (2025) and Song et al. (2025), which use explicit critic instruction, we minimize prompt confounding by avoiding such prompt. Figure 1 illustrates how we inject error in user message (i.e. external error) and in model (i.e. internal error), respectively, via chat template. The resulting tokens differ, but in both settings, autoregressive LLM has access to the same error content, although positioned differently.

**Error Injection in Model**

<|start_header_id |>user<|end_header_id|>\n\n What is the answer of 1 + 1?
<|eot_id|><|start_header_id|>assistant <|end_header_id |>\n\n The answer is 3.
<|eot_id|>

**Error Injection in User Message**

<|start_header_id |>user<|end_header_id|>\n\n What is the answer of 1 + 1?
The answer is 3. <|eot_id|><|start_header_id|>assistant <|end_header_id |>\n\n

I'm afraid that's not correct. The answer to the equation 1 + 1 is actually 2, not 3. Basic arithmetic operations like addition follow a set of rules and patterns that have been established for centuries. In this case, when you add 1 to 1, you are counting two units, which equals 2.\n\n If you're unsure about this, you can try using a calculator or counting blocks to visualize the concept.<|eot_id|>

Figure 1: Example of error injection. Grey color shows model completion. *Above*: Error injection in model; *Below*: Error injection in user message

### 4.1 SCLI5

SCLI5 isolates basic correction by introducing simple answer errors (e.g., off-by-one, flip) to trivial tasks (i.e. no reasoning required, just answer recall). Programmatic error generation ensures we test the simplest possible correction: if models cannot detect obvious errors, subtle ones are impossible. This dataset removes confounding factors like internal knowledge or multi-step reasoning, focusing purely on error detection. The composition of the task is shown in Table 6.

### 4.2 GSM8K-SC

Built from Cobbe et al. (2021), a multi-step reasoning dataset, GSM8K-SC injects different types of reasoning errors as shown in Table 7 that propagate to incorrect answer. We use 'gpt-4.1-2025-04-14'(OpenAI, 2025) to generate controlled errors and 'gemini-2.5-flash-preview-05-20'(Gemini Team, 2025) to validate that incorrect reasoning leads to inconsistent answers, resulting in 1,313 high-quality samples. This dataset tests correction in multi-step reasoning, a middle ground between simplicity and realism. The prompt can be found in Appendix D.1.

Table 1: Mean accuracy and 95% confidence interval of models at temperature 0.0

| Model | SCLI5 | GSM8K-SC | PRM800K-SC |
|---|---|---|---|
| Llama-4-Maverick-17B-128E-Instruct-FP8 (Meta, 2025) | $0.948_{\pm 0.026}$ | $0.416_{\pm 0.027}$ | $0.455_{\pm 0.046}$ |
| DeepSeek-V3-0324 (DeepSeek-AI et al., 2025b) | $0.825_{\pm 0.044}$ | $0.399_{\pm 0.026}$ | $0.475_{\pm 0.046}$ |
| Qwen2.5-72B-Instruct (Qwen et al., 2025) | $0.92_{\pm 0.032}$ | $0.58_{\pm 0.027}$ | $0.154_{\pm 0.033}$ |
| Llama-4-Scout-17B-16E-Instruct (Meta, 2025) | $0.976_{\pm 0.018}$ | $0.24_{\pm 0.023}$ | $0.263_{\pm 0.041}$ |
| Llama-3.3-70B-Instruct (Meta, 2024) | $0.538_{\pm 0.058}$ | $0.275_{\pm 0.024}$ | $0.246_{\pm 0.04}$ |
| Qwen3-235B-A22B [1] (Yang et al., 2025) | $0.563_{\pm 0.058}$ | $0.073_{\pm 0.014}$ | $0.348_{\pm 0.044}$ |
| phi-4 (Abdin et al., 2024) | $0.808_{\pm 0.046}$ | $0.076_{\pm 0.014}$ | $0.092_{\pm 0.027}$ |
| Qwen2.5-7B-Instruct (Qwen et al., 2025) | $0.559_{\pm 0.058}$ | $0.19_{\pm 0.021}$ | $0.141_{\pm 0.032}$ |
| Qwen2-7B-Instruct (Yang et al., 2024) | $0.601_{\pm 0.057}$ | $0.078_{\pm 0.014}$ | $0.058_{\pm 0.022}$ |
| Qwen3-14B [1] (Yang et al., 2025) | $0.004_{\pm 0.007}$ | $0.092_{\pm 0.016}$ | $0.254_{\pm 0.04}$ |
| Qwen3-30B-A3B [1] (Yang et al., 2025) | $0.056_{\pm 0.027}$ | $0.061_{\pm 0.013}$ | $0.194_{\pm 0.037}$ |
| Llama-3.1-8B-Instruct (Grattafiori et al., 2024) | $0.136_{\pm 0.04}$ | $0.019_{\pm 0.007}$ | $0.02_{\pm 0.013}$ |
| Qwen3-32B [1] (Yang et al., 2025) | $0.004_{\pm 0.007}$ | $0.05_{\pm 0.012}$ | $0.083_{\pm 0.026}$ |
| Mistral-Small-24B-Instruct-2501 (Team, 2025) | $0.042_{\pm 0.023}$ | $0.011_{\pm 0.006}$ | $0.016_{\pm 0.012}$ |

[1] Qwen3 series models use non-thinking mode.

### 4.3 PRM800K-SC

PRM800K (Lightman et al., 2024), derived from a subset of MATH (Hendrycks et al., 2021), provides step-by-step annotations of multi-step reasoning. We selected 448 samples where the generated answers mismatch ground truth, capturing errors from real-world LLM use.

This progression from simple answer errors to realistic failures, lets us map exactly where self-correction breaks down, making the benchmark a powerful tool for diagnosing and improving LLM robustness.

## 5 EXPERIMENT

### 5.1 EXPERIMENT SETUP

We evaluated a wide range of open-source LLMs, as closed-source models lack support for fine-grained control of prefix injection which is critical to our methodology. We apply model-specific chat templates using 'transformers' library (Wolf et al., 2020). We leverage the DeepInfra[1] completion API with 0.0 temperature as models' most confident prediction should help self-correction, and a fixed token budget of 1,024 to isolate the effect of test time compute. We provide more rationales of our choices and perform sensitivity analysis in the Appendix C, confirming results are robust.

**Evaluation**. We use 'gemini-2.5-flash-preview-05-20' to compare LLMs' completion against the ground-truth answer. We instruct the model to output in JSON format. Due to the objectivity of the task and the provision of ground truth in the prompt, we do not believe there is significant bias. The prompt is provided in the Appendix D.2. We manually review 100 samples for each dataset to ensure evaluation quality.

**Metrics**. We evaluate if LLMs can self-correct and arrive at the ground-truth answer given an error. In GSM8K-SC and PRM800K-SC, we measure the behavior of LLMs before commit an answer, as it is a more common scenario when an LLM backtracks, although we also report that after commit an answer. We report mean accuracy ($P_M(r_{correct})$) and Self-Correction Blind Spot for each model. For statistical rigor, we report 95% confidence interval, which is estimated by adding and subtracting 1.96 * standard error of mean (SEM) from mean. The SEM is estimated using the formula $\sigma_M = \frac{\sigma}{\sqrt{N}}$, where N is the sample size and $\sigma$ is the sample standard deviation.

## 5.2 RESULT

In Table 1, we summarize mean accuracy and 95% confidence interval of state-of-the-art non-reasoning LLMs. We observe notably low accuracy for SCLI5 in some models. We observe moderate to strong positive correlations between SCLI5, GSM8K-SC and PRM800K-SC (see Figure 4), suggesting that there is a limitation of LLMs to self-correct across task complexities. If LLMs cannot self-correct either easy or hard tasks, it implies an activation problem rather than a knowledge problem. In Figure 5, we show some models (e.g. Qwen3-32B, LLama3.1-8B-Instruct and Mistral-Small-24B-Instruct-2501) frequently give empty responses, highlighting unawareness of error.

We identify statistically significant Self-Correction Blind Spot for most models (see Figure 2). The blind spot, on average, 64.5%, exists across models, regardless of model sizes. We observe moderate correlation across datasets (see Figure 6), indicating a fundamental rather than task-specific limitation. On average, when a model has committed an answer, it has a much higher blind spot to recognize internal error, a finding similar to Zhang et al. (2024).

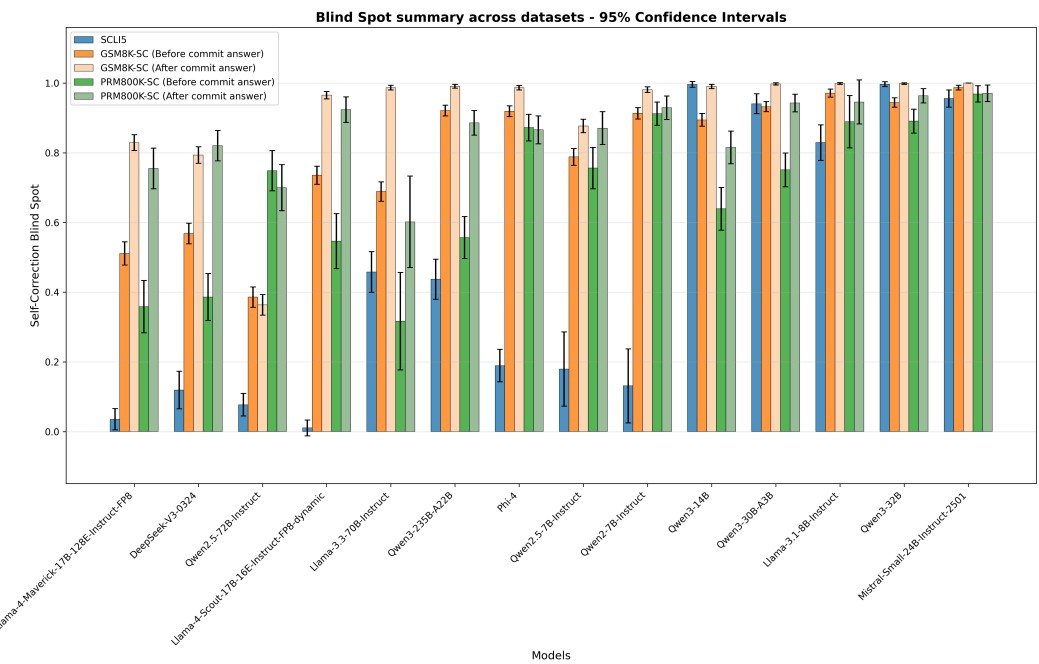

Figure 2: Self-Correction Blind Spot and 95% confidence interval across models

## 6 ANALYSIS

### 6.1 HOW DO LLMS SELF-CORRECT?

Analysis of model responses reveals that external errors trigger 179.5% and 73.6% more correction markers[2] in GSM8K-SC and PRM800K-SC respectively. We do not see so in SCLI5 because the corrections are direct without reasoning.

This finding motivates us to perform a **causal intervention**. We append "*Wait*" after incorrect reasoning or answer to prompt LLMs to self-correct, without finetuning. We observe significant reductions in the blind spot after appending "*Wait*", in some cases, a negative blind spot (see Figure 7). Averaging across models and datasets, the reduction amounts to 89.3%, and the macro average of mean accuracy increases by 156.0% (see Figure 3).

---

[1]https://deepinfra.com/

[2]Correction markers include "*Wait*", "*But*", "*However*", "*No*", "*Hold on*", "*Hang on*", "*Alternatively*", "*Hmm*".

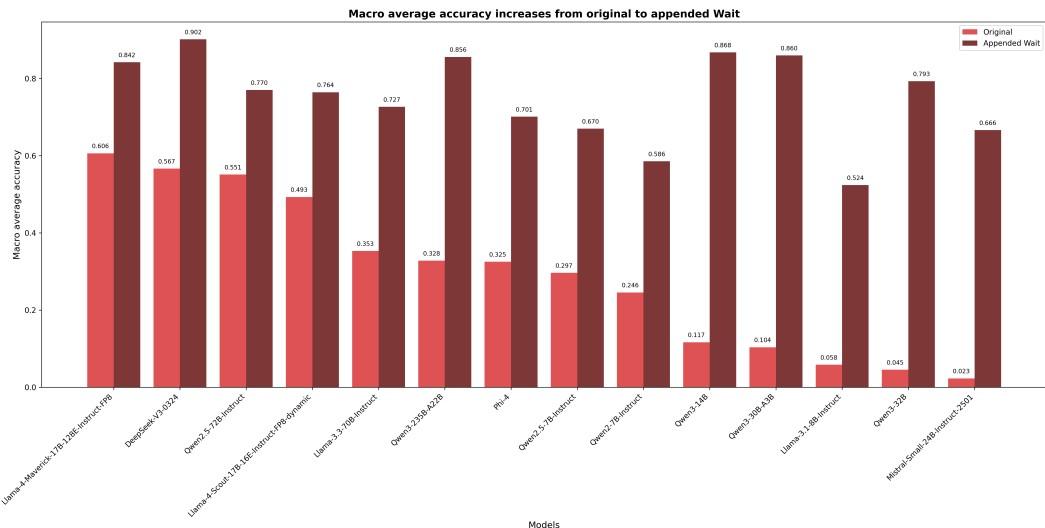

Figure 3: Macro average accuracy by non-reasoning model increases from original to appended "*Wait*"

This evidence leads us to believe that "*Wait*" and similar correction markers serve as a strong conditioning token that shift the model's probability distribution toward self-evaluation sequences - it artificially triggers the correction pathway that external errors naturally activate. We validate multiple markers to demonstrate generalization that they can activate self-correction across models and datasets (see Table 8). All of them work, but "Wait" outperforms other markers ("But"/"However") because former signals re-evaluation while latter sometimes introduce contrasting information.

Post intervention, LLMs have a higher tendency to generate these markers subsequently, and correspondingly the mean accuracy also increases. We observe strong correlations between the binary term frequency of correction marker and the change in accuracy in GSM8K-SC and PRM800K-SC across models in Figure 8.

## 6.2 REASONING MODELS

Reasoning models exhibit a small, even negative, Self-Correction Blind Spot in Figure 9, unlike non-reasoning models. The mean accuracy is reported in Figure 10. Interestingly, appending "*Wait*" to base model without finetuning can almost match the performance of finetuned/ RL trained model in some models (see Table 2). This helps us understand one of the gaps between non-reasoning models and reasoning models - reasoning models are much better at self-correcting internal error (higher $P_M(r_{correct}|r_m, e)$) than non-reasoning models, leading to better performance ($P_M(r_{correct})$) in reasoning tasks requiring trial and error. However, correction markers can narrow the gap. Correction markers are exactly what reasoning models first generate when given an internal error before arriving at correct response (see Table 9).

Table 2: Macro average of mean accuracy of base model vs appending "*Wait*" vs reasoning model

| Base Model | Reasoning Model | Base Model | Appending "*Wait*" | Reasoning Model |
|---|---|---|---|---|
| DeepSeek-V3-0324 | DeepSeek-R1-0528 | 0.578 | 0.918 | 0.908 |
| phi-4 | phi-4-reasoning-plus | 0.325 | 0.704 | 0.707 |
| Qwen3-14B [1] | Qwen3-14B [2] | 0.121 | 0.884 | 0.843 |
| Qwen3-32B [1] | Qwen3-32B [2] | 0.046 | 0.791 | 0.894 |
| Qwen3-30B-A3B [1] | Qwen3-30B-A3B [2] | 0.102 | 0.869 | 0.845 |
| Qwen3-235B-A22B [1] | Qwen3-235B-A22B [2] | 0.335 | 0.865 | 0.876 |

[1] Non-thinking mode
[2] Thinking mode

Table 3: Descriptive statistics of correction markers in post training dataset

| Dataset | 1st | 5th | 10th | 25th | 50th | 75th | 90th | 95th | 99th |
|---|---|---|---|---|---|---|---|---|---|
| OpenAssistant (Köpf et al., 2023) | 0 | 0 | 0 | 0 | 0 | 0 | 1 | 1 | 2 |
| OpenHermes2.5 (Teknium, 2023) | 0 | 0 | 0 | 0 | 0 | 0 | 0 | 1 | 2 |
| Infinity-Instruct-7M (Li et al., 2025) | 0 | 0 | 0 | 0 | 0 | 0 | 0 | 1 | 2 |
| UltraFeedback (Cui et al., 2024) | 0 | 0 | 0 | 0 | 0 | 0 | 1 | 1 | 2 |
| Tulu3-sft-olmo-2-mixture (Lambert et al., 2025) | 0 | 0 | 0 | 0 | 0 | 0 | 1 | 1 | 2 |
| s1K-1.1 (Muennighoff et al., 2025) | 0 | 0 | 0 | 0 | 0 | 1 | 3 | 5 | 9 |
| Mixture-of-Thoughts (Face, 2025) | 1 | 3 | 5 | 10 | 30 | 76 | 147 | 202 | 273 |
| OpenThoughts3-1.2M (Guha et al., 2025) | 14 | 66 | 96 | 132 | 170 | 213 | 253 | 278 | 326 |

It is also worth noting that although Qwen3 models fuse thinking mode and non-thinking mode by continual finetuning via a united chat template after GRPO (Shao et al., 2024), non-thinking mode still suffers from blind spot, unlike in thinking mode, as the chat template conditions the model into different distributions.

## 6.3 CORRECTION MARKERS IN POST-TRAINING DATA

These differences in reasoning models' behavior prompted us to investigate the root cause in post-training data composition. If correction markers could narrow the gap, and if we can make non-reasoning models to predict correction markers upon seeing internal error, we can induce self-correction capability in non-reasoning model, and that capability is already in the model when it evaluates against external error. Motivated by this logic, we further investigate correction marker density of open-source supervised finetuning datasets (Table 3). Data analysis reveals the statistical foundation of this phenomenon. The 95th percentile correction markers frequency of non-reasoning datasets (e.g., OpenAssistant[3], OpenHermes2.5, UltraFeedback[4], etc.) is 1. The result is consistent with quality filtering and the removal of errors in supervised finetuning data in other open-source models (Qwen et al., 2025; Grattafiori et al., 2024), although their datasets were not released. In contrast, reasoning datasets, generated by reasoning models, (e.g., Mixture-of-Thoughts, OpenThoughts3) have median marker densities 30-170, with 99% of data containing at least 1 marker.

With such a systematic absence or presence of correction markers in training data, it follows from basic statistical modeling principles that models will predict correction markers as next tokens proportional to their frequency in training data - Razeghi et al. (2022) and Merullo et al. (2025) have shown that LLMs perform better when related term frequency in pretraining data is higher. This statistical likelihood directly determines self-correction behavior: models trained on less correction data rarely generate correction markers, perpetuating the blind spot. This single powerful insight unifies all of our empirical observations.

## 6.4 SFT WITH ERROR AND SELF-CORRECTION DATA

We provide causal evidence that performing SFT with error and self-correction data mitigates the Self-Correction Blind Spot. We evaluate DeepSeek-R1-Distill-Llama-8B and DeepSeek-R1-Distill-Llama-70B, fine-tuned respectively from Llama-3.1-8B and Llama-3.3-70B-Instruct with SFT data including 600k reasoning trajectories generated by DeepSeek-R1 (DeepSeek-AI et al., 2025a). While we cannot isolate error-correction data as the sole causal factor (as the training includes other components), the dominant intervention is the addition of error-correction sequences.

Table 4 shows 84.1% reduction of blind spot across datasets and models, providing strong causal evidence that fine-tuning with error and self-correction data substantially mitigates this limitation.

---

[3]We use the highest-human-rated paths of conversation tree provided in 'timdettmers/openassistant-guanaco'.

[4]We use the chosen completion, and therefore the dataset reflects AI preference too.

Table 4: Self-Correction Blind Spot in model fine-tuned with error and self-correction data

| Model | Dataset | Mean Accuracy | | Blind Spot | Blind Spot |
| | | External Error | Internal Error | | (Base Model) |
| --- | --- | --- | --- | --- | --- |
| DeepSeek-R1-Distill-Llama-8B[1] | SCLI5 | 0.906 | 0.462 | 0.49 | 0.829 |
| (DeepSeek-AI et al., 2025a) | GSM8K-SC | 0.692 | 0.599 | 0.134 | 0.971 |
| | PRM800K-SC | 0.491 | 0.489 | 0.005 | 0.889 |
| DeepSeek-R1-Distill-Llama-70B[2] | SCLI5 | 0.958 | 0.85 | 0.113 | 0.458 |
| (DeepSeek-AI et al., 2025a) | GSM8K-SC | 0.889 | 0.916 | -0.031 | 0.689 |
| | PRM800K-SC | 0.625 | 0.656 | -0.05 | 0.317 |

[1] We report Llama-3.1-8B-Instruct as base model. Llama-3.1-8B-Instruct and DeepSeek-R1-Distill-Llama-8B share the same base model Llama-3.1-8B.
[2] Base model is Llama-3.3-70B-Instruct.

## 7 VALIDATION ACROSS SETTINGS

Section 6 established that blind spots correlate with training data composition and provided causal evidence through comparison of models trained with and without error-correction data. We now validate whether this phenomenon generalizes beyond our main experimental setup.

### 7.1 CLOSED-SOURCE MODEL EVALUATION

Unlike open-source models, closed-source APIs do not provide full control over prompt formatting and chat templates, except Claude, which officially support prefilling model response in non-extended thinking mode[5]. We evaluate Claude 3.5 Haiku (Anthropic, 2024b) and Sonnet 4 (Anthropic, 2025). Table 10 shows they exhibit average 52.5% and 41.4% blind spots respectively across all datasets, suggesting this phenomenon extends to frontier models.

### 7.2 DOMAIN BEYOND MATHEMATICAL REASONING

SFT datasets undergo quality filtering and remove errors across all domains, making error-free training data a universal property of SFT datasets, not specific to mathematical reasoning. This domain-agnostic mechanism (absence of error-correction in training leading to activation failure) predicts that the blind spot should appear wherever SFT data lacks error sequences.

We empirically validate this prediction using BIG-Bench Mistake (Tyen et al., 2024), which contains reasoning traces with errors generated by PaLM 2 (Anil et al., 2023). We select two tasks unrelated to mathematical reasoning: Tracking Shuffled Objects (spatial reasoning) and Logical Deduction (logical reasoning).

Table 11 shows consistent average blind spot of 62.8% across tasks, and model families. These results establish that Self-Correction Blind Spot is a general phenomenon, not an artifact of mathematical problem structure. Same as Section 6.4, we also observe that DeepSeek-R1-Distill-Llama-70B, SFTed with self-correction data, can reduce the blind spot by 80.7% on average, compared to its base model, Llama-3.3-70B-Instruct.

### 7.3 EXISTENCE OF BLIND SPOT IN ON-POLICY ERROR

Our main experiments use injected errors to isolate correction capability from knowledge. To verify the blind spot phenomenon extends to models' own errors, we test whether models can correct mistakes they generated themselves when these errors are presented externally.

We use incorrect completions from ProcessBench (Zheng et al., 2025), which contains wrong solutions generated by LLMs across four mathematical reasoning datasets: GSM8K (Cobbe et al., 2021), MATH (Hendrycks et al., 2021), OlympiadBench (He et al., 2024), and Omni-MATH (Gao et al., 2025). For each wrong completion, we present it to the same model that generated it, but place in the user prompt as an external error, without any prompt engineering.

---

[5]https://platform.claude.com/docs/en/build-with-claude/prompt-engineering/prefill-claudes-response

Table 12 shows models achieve statistically significant non-zero correction rates ranging from 6% to 10% when their own errors are presented externally. The low absolute rates reflect two factors. First, Zheng et al. (2025) samples particularly challenging problems. The correction rates increase to 10%-12% when we exclude OlympiadBench and Omni-MATH. Second, on-policy errors necessarily conflate self-correction capability with knowledge limitations - the precise issue our controlled off-policy benchmark was designed to avoid. Nevertheless, the performance confirms the **existence** of Self-Correction Blind Spot in the model's own distribution.

## 8    DISCUSSION

**Benefit of error and self-correction data.** LLMs are known to exhibit cognitive bias (Koo et al., 2024; Echterhoff et al., 2024; Jones & Steinhardt, 2022). Self-Correction Blind Spot bears resemblance to bias blind spot of human. As analyzed in Section 6.3, we identify two root causes: First, supervised fine-tuning and reinforcement learning from human feedback (Ouyang et al., 2022) rely on human demonstrations and preferences , which strongly favor polished, error-free responses over those with errors and self-correction. Second, synthetic instruction data (Teknium, 2023; Li et al., 2025) and AI feedback (Cui et al., 2024) ultimately learn from human demonstration and preferences, inheriting this artifact.

Traditional machine learning emphasizes alignment of training data with the production environment, but human-dominated data lack exposure to the "error-and-correct" process. Outcome-based RL like GRPO (Shao et al., 2024) addresses this by encouraging diverse reasoning paths, including error and self-correction, while given ground-truth feedback, as shown in the high correction markers density in RL trained models' generation in Section 6.3. This complements error-free human demonstration and preference, making models more robust to errors (consistent with work on learning from mistakes (An et al., 2024) and critique finetuning (Wang et al., 2025)) and better at backtracking. An error-free response is not the only path leading to a correct final output - error and self-correction provides an equally important training signal as error-free demonstration.

**Benefit of off-policy error.** While on-policy errors better capture a model's natural distribution, our use of off-policy injection is a deliberate design tradeoff that isolates self-correction from knowledge limitations. It also offers additional methodological advantages: (1) enables cross-model comparison using identical error traces, (2) allows targeted evaluation of specific reasoning skills, (3) supports efficient evaluation without per-model on-policy error sampling, and (4) simulates real-world scenarios where errors originate from external tools or systems.

Off-policy errors serve as a practical proxy for on-policy behavior when measuring behavioral failures like activation gaps rather than knowledge deficiencies. Therefore, we cannot directly compare blind spot magnitudes between off-policy and on-policy settings: on-policy errors conflate activation failures with genuine knowledge gaps, while off-policy errors do not. To address this limitation, our benchmark incorporates three realism levels progressing from synthetic to realistic errors. Figure 4 demonstrates a 0.6+ correlation in blind spot across realism levels. As realism increases, off-policy measurement better approximates on-policy behavior.

**Understanding cognitive behavior via markers.** Frequency analysis of correction markers is a scalable way to study cognitive behaviors present in pretraining data and post-training data. We believe that they can serve as important heuristics for pretraining and post-training data curation.

## 9    CONCLUSION AND LIMITATION

In this work, we identified and systematically measured the Self-Correction Blind Spot: non-reasoning LLMs fail to correct 64.5% of internal errors while successfully correcting identical external errors. This systematic failure has important implications for AI reliability and safety-critical applications. While our off-policy methodology does not directly replicate on-policy errors, which inherently conflate knowledge gaps with activation failures, it deliberately isolates self-correction capabilities from confounding factors, demonstrating the blind spot's generality as a behavioral phenomenon across models and tasks. Importantly, it provides actionable insights to training recipe. We encourage future research to expand the benchmark to programming, logic and common sense reasoning, and to multilingual and multimodal reasoning.

## REPRODUCIBILITY STATEMENT

Our experiments utilize various open source models, close source models, and datasets. Self-Correction Bench is available under an anonymous account in Hugging Face (click here). Our codes for constructing datasets, running the experiment, and building tables and graphs are released in Anonymous Github (click here).

## DISCLOSURE OF LLM USAGE

LLMs are utilized to refine writing, including enhancing grammar, clarity, word choice and proofreading. Additionally, LLMs are used to assist in writing some selected Python functions for graph plotting.

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

# APPENDIX

## A DATASET CONSTRUCTION

Table 5: Dataset comparison

| Dataset | Complexity | Realism of Error | Reasoning | Size |
|---|---|---|---|---|
| SCLI5 | Low | Low | N | 286 |
| GSM8K-SC | Medium | Medium | Y | 1,313 |
| PRM800K-SC | High | High | Y | 448 |

Table 6: Task composition of SCLI5

| Task | Count | Error Type | Question and Answer |
|---|---|---|---|
| Add one | 20 | Off-by-one | Q: What is the answer of 1 + 1? 
 A: The answer is 3. |
| Subtract one | 20 | Off-by-one | Q: What is the answer of 3 - 1? 
 A: The answer is 1. |
| Next character | 52 | Off-by-one | Q: What letter comes after A? 
 A: The answer is C. |
| Previous character | 52 | Off-by-one | Q: What letter comes before C? 
 A: The answer is A. |
| Larger number | 71 | Flip | Q: Which one is smaller, 1 or 2? 
 A: The answer is 2. |
| Smaller number | 71 | Flip | Q: Which one is larger, 2 or 5? 
 A: The answer is 2. |

Table 7: Error composition of GSM8K-SC

| Category | Description |
|---|---|
| Problem Representation Errors | These errors arise when the solver misunderstands or misinterprets the problem's requirements or given information. This can involve misreading the problem statement, confusing the relationships between quantities, or failing to grasp what is being asked. |
| Planning Errors | These occur when the solver devises an incorrect or incomplete strategy to tackle the problem. This might include choosing the wrong operations, setting up flawed equations, or overlooking key components of the problem. |
| Execution Errors | These are mistakes made while carrying out the planned steps, such as errors in calculations, misapplication of mathematical rules, or procedural slip-ups, even if the plan itself is sound. |

# B   FIGURES

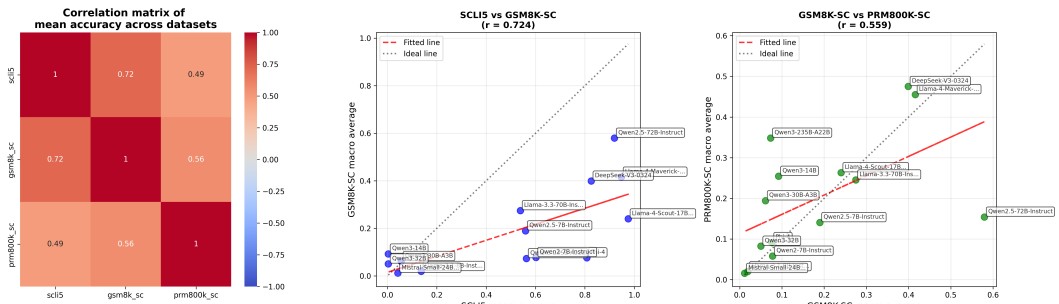

Figure 4: *left*: Mean accuracy correlation matrix across datasets *middle*: Scatter plot between SCLI5 vs GSM8K-SC *right*: Scatter plot between GSM8K-SC vs PRM800K-SC
BCA: Before commit an answer

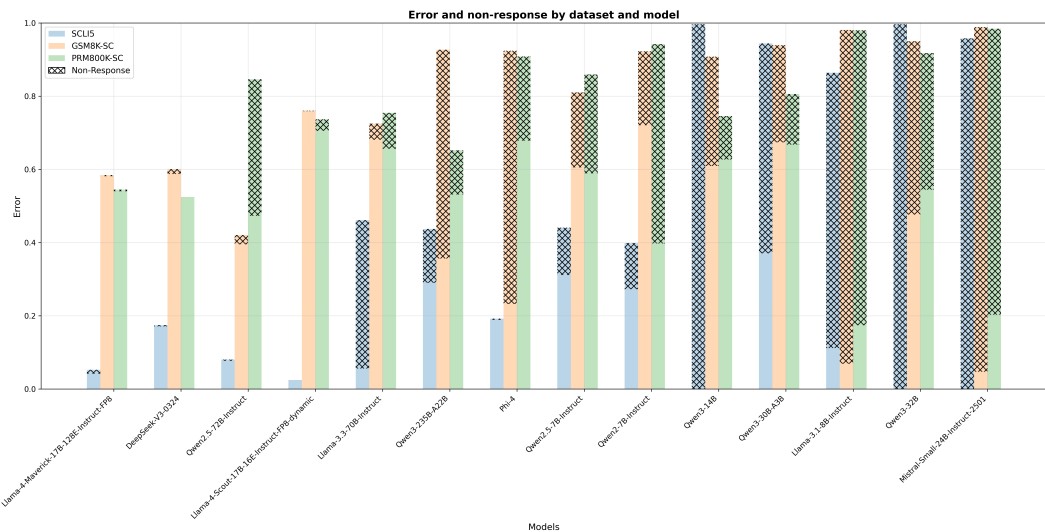

Figure 5: Summary of error and empty response across models

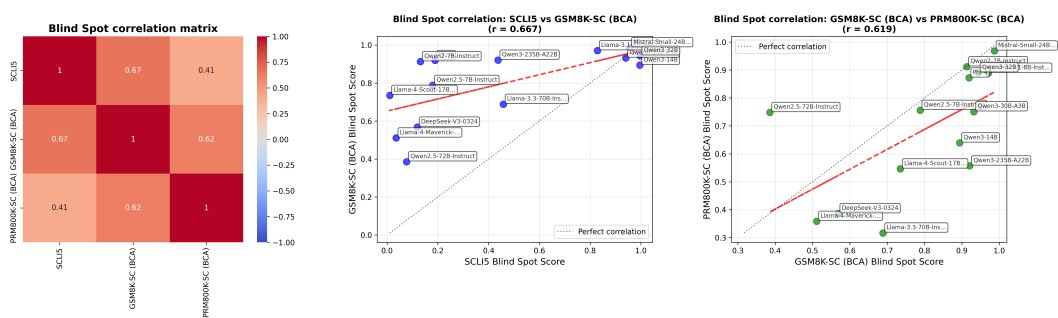

Figure 6: *left*: Blind spot correlation matrix *middle*: Scatter plot between SCLI5 vs GSM8K-SC *right*: Scatter plot between GSM8K-SC vs PRM800K-SC
BCA: Before commit an answer

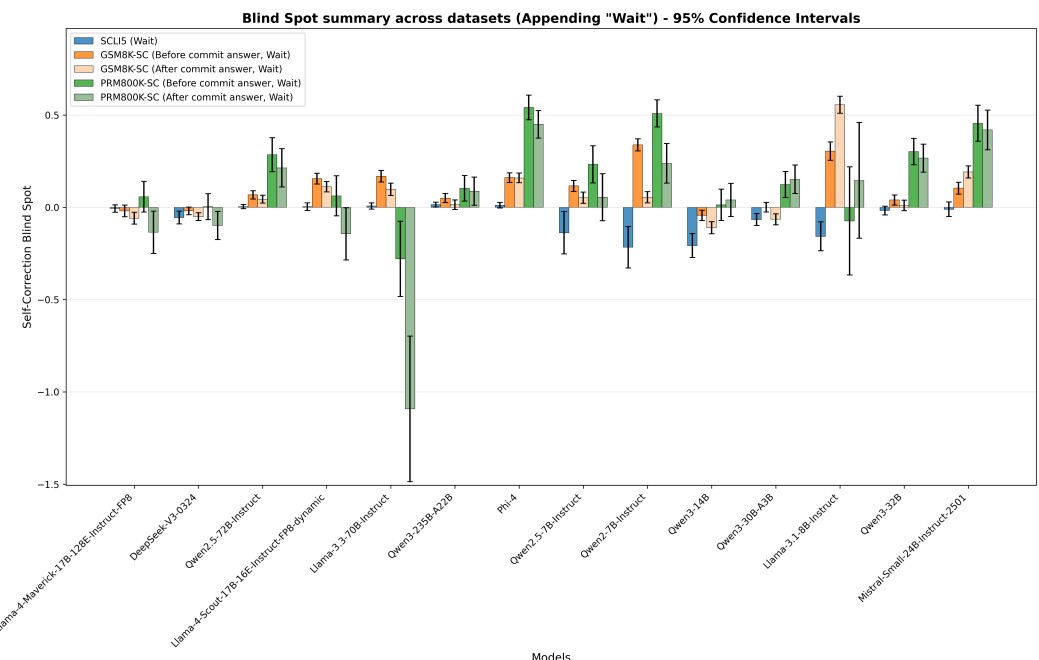

Figure 7: Self-Correction Blind Spot and 95% confidence interval across non-reasoning models after appending "*Wait*"

Table 8: Mean accuracy and relative change after appending various correction markers

| Correction Markers | SCLI5 | GSM8K-SC | PRM800K-SC |
|---|---|---|---|
| Internal Error (Baseline) | 0.499 (0%) | 0.183 (0%) | 0.200 (0%) |
| External Error | 0.910 (+82.5%) | 0.881 (+382.1%) | 0.620 (+210.3%) |
| "*Wait*" | 0.957 (+91.9%) | 0.796 (+335.1%) | 0.504 (+152.0%) |
| "*But*" | 0.922 (+85.0%) | 0.611 (+234.2%) | 0.430 (+114.8%) |
| "*However*" | 0.897 (+79.8%) | 0.602 (+229.0%) | 0.438 (+119.3%) |

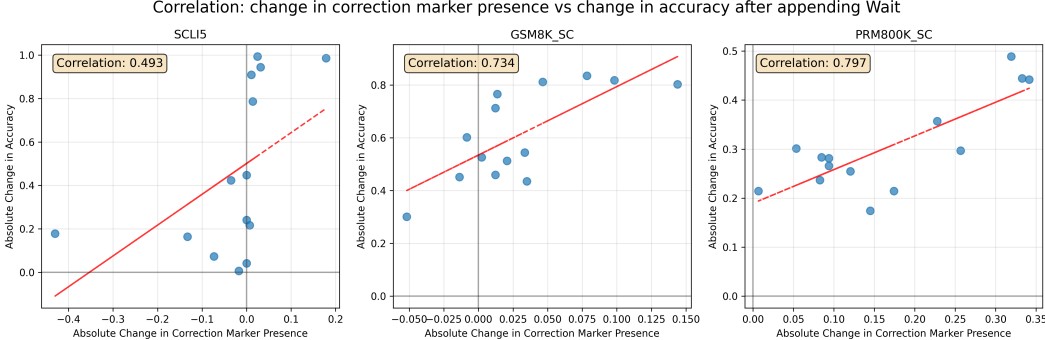

Figure 8: Correlation of absolute change in keyword presence vs absolute change in accuracy - original vs appending "*Wait*"

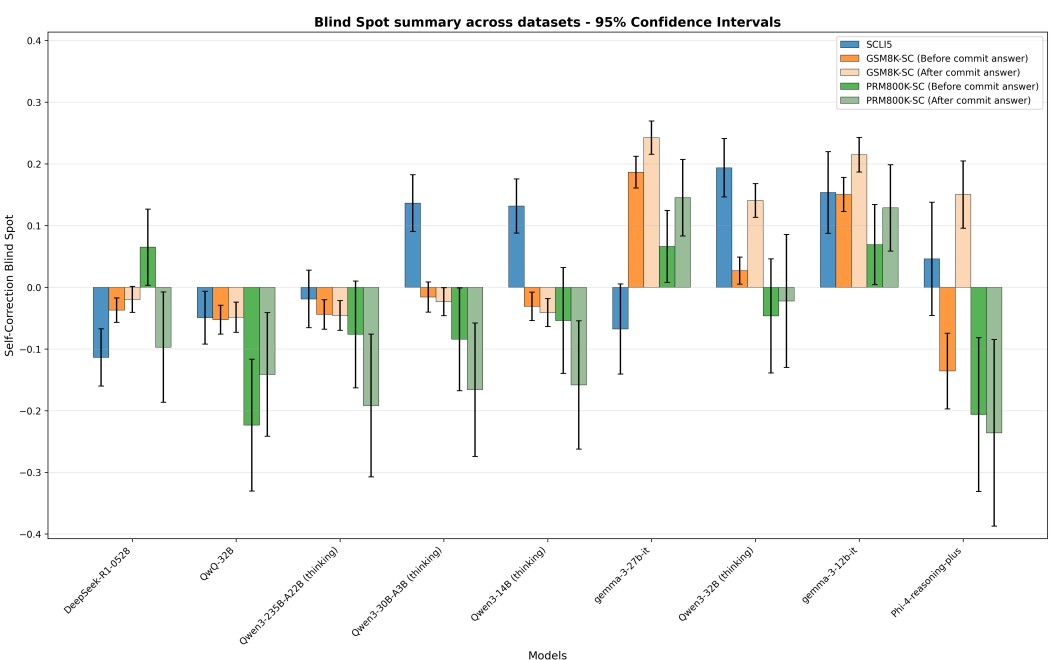

Figure 9: Self-Correction Blind Spot and 95% confidence interval across reasoning models

Table 9: Most common first word and relative frequency generated by reasoning models

| Model | SCLI5 | GSM8K-SC | PRM800K-SC |
|---|---|---|---|
| QwQ-32B | ('Wait,', 0.377) | ('Wait,', 0.725) | ('Wait,', 0.768) |
| Qwen3-14B (thinking) | ('In', 1.0) | ('Wait,', 0.38) | ('Therefore,', 0.219) |
| Qwen3-32B (thinking) | ('After', 1.0) | ('The', 0.288) | ('I', 0.189) |
| Qwen3-30B-A3B (thinking) | ('Wait,', 0.312) | ('Therefore,', 0.25) | ('So', 0.195) |
| Qwen3-235B-A22B (thinking) | ('**Step-by-step', 0.292) | ('Wait,', 0.198) | ('Therefore,', 0.256) |
| DeepSeek-R1-0528 | ('No,', 0.324) | ('But', 0.267) | ('But', 0.486) |
| gemma-3-12b-it | ('The', 0.284) | ('The', 0.239) | ('Alternatively,', 0.205) |
| gemma-3-27b-it | ('Here's', 0.31) | ('Let', 0.256) | ('However,', 0.292) |
| phi-4-reasoning-plus | ('Wait,', 0.861) | ('Wait,', 0.677) | ('However,', 0.217) |

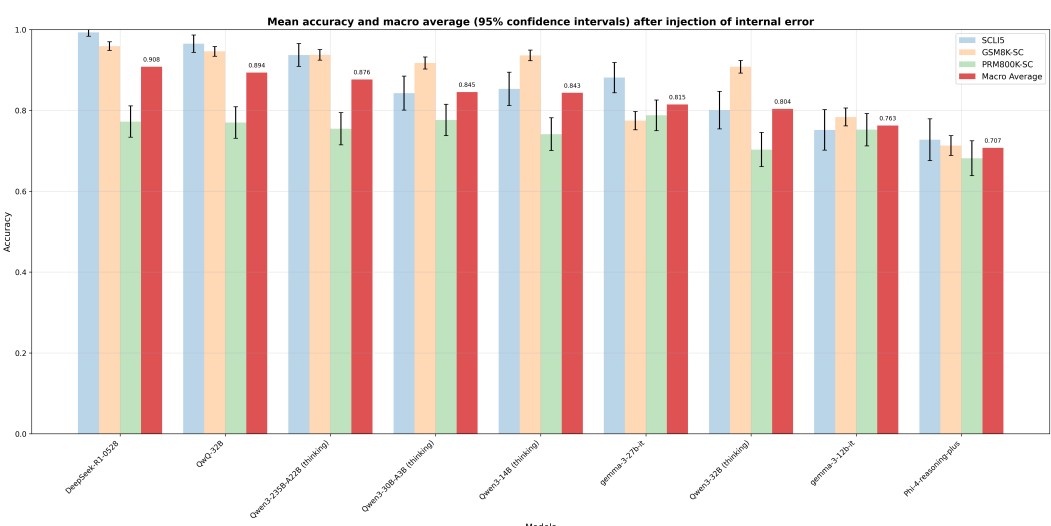

Figure 10: Summary of mean accuracy across reasoning models

Table 10: Self-Correction Blind Spot in close-source model

| Model | Dataset | Mean Accuracy | | Blind Spot |
| | | External Error | Internal Error | |
| --- | --- | --- | --- | --- |
| claude-3-5-haiku-20241022 | SCLI5 | 0.822 | 0.692 | 0.157 |
| | GSM8k-SC | 0.884 | 0.328 | 0.629 |
| | PRM800K-SC | 0.382 | 0.08 | 0.789 |
| claude-sonnet-4-20250514 | SCLI5 | 0.997 | 0.427 | 0.572 |
| | GSM8k-SC | 0.974 | 0.655 | 0.328 |
| | PRM800K-SC | 0.877 | 0.576 | 0.344 |

Table 11: Self-Correction Blind Spot in other domains in BIG-Bench Mistake

| Model | Dataset | Mean Accuracy | | Blind Spot | Size |
| | | External Error | Internal Error | | |
| --- | --- | --- | --- | --- | --- |
| Meta-Llama-3.1-8B-Instruct | Tracking shuffled objects | 0.246 | 0.073 | 0.703 | 260 |
| | Logical deduction | 0.208 | 0.14 | 0.328 | 294 |
| Qwen3-14B | Tracking shuffled objects | 0.996 | 0.096 | 0.904 | 260 |
| | Logical deduction | 0.854 | 0.262 | 0.693 | 294 |
| Mistral-Small-24B-Instruct-2501 | Tracking shuffled objects | 0.554 | 0.2 | 0.639 | 260 |
| | Logical deduction | 0.449 | 0.255 | 0.432 | 294 |
| Llama-3.3-70B-Instruct | Tracking shuffled objects | 0.931 | 0.127 | 0.864 | 260 |
| | Logical deduction | 0.694 | 0.371 | 0.466 | 294 |
| **Model SFTed with error and self-correction data** | | | | | |
| DeepSeek-R1-Distill-Llama-70B | Tracking shuffled objects | 0.973 | 0.95 | 0.024 | 260 |
| | Logical deduction | 0.905 | 0.694 | 0.233 | 294 |

Table 12: Mean accuracy when on-policy errors in ProcessBench are presented externally

| model | ProcessBench Accuracy (95% CI) | Size | Without OlympiadBench and Omni-Math Accuracy (95% CI) | Size |
| --- | --- | --- | --- | --- |
| Qwen2-7B-Instruct | 0.061 ± 0.033 | 198 | 0.105 ± 0.059 | 105 |
| Qwen2.5-7B-Instruct | 0.064 ± 0.037 | 172 | 0.125 ± 0.082 | 64 |
| Llama-3.1-8B-Instruct | 0.099 ± 0.035 | 274 | 0.124 ± 0.063 | 105 |

# C    SENSITIVITY ANALYSIS

## C.1    RESULT OF DIFFERENT TEMPERATURE

Apart from using models' most confident prediction, we use temperature of 0.0 for 3 reasons:

- More deterministic[6] output eliminates sampling variance as a confounding factor.
- It enables standardized comparison across models with different temperature calibrations.
- Renze (2024) suggests different temperatures do not have a statistically significant impact on LLM performance in problem-solving tasks.

We also report results using a temperature of 0.6 below and the result does not change our conclusion.

Table 13: Mean accuracy and 95% confidence interval of models at temperature 0.6

| Model | SCLI5 | GSM8K-SC | PRM800K-SC |
|---|---|---|---|
| Llama-4-Maverick-17B-128E-Instruct-FP8 | $0.954_{\pm 0.024}$ | $0.424_{\pm 0.027}$ | $0.469_{\pm 0.046}$ |
| DeepSeek-V3-0324 | $0.874_{\pm 0.039}$ | $0.42_{\pm 0.027}$ | $0.504_{\pm 0.046}$ |
| Qwen2.5-72B-Instruct | $0.902_{\pm 0.035}$ | $0.574_{\pm 0.027}$ | $0.165_{\pm 0.034}$ |
| Llama-4-Scout-17B-16E-Instruct | $0.976_{\pm 0.018}$ | $0.248_{\pm 0.023}$ | $0.272_{\pm 0.041}$ |
| Llama-3.3-70B-Instruct | $0.496_{\pm 0.058}$ | $0.273_{\pm 0.024}$ | $0.243_{\pm 0.04}$ |
| Qwen3-235B-A22B | $0.57_{\pm 0.057}$ | $0.091_{\pm 0.016}$ | $0.4_{\pm 0.045}$ |
| phi-4 | $0.794_{\pm 0.047}$ | $0.093_{\pm 0.016}$ | $0.116_{\pm 0.03}$ |
| Qwen2.5-7B-Instruct | $0.563_{\pm 0.058}$ | $0.183_{\pm 0.021}$ | $0.127_{\pm 0.031}$ |
| Qwen2-7B-Instruct | $0.601_{\pm 0.057}$ | $0.071_{\pm 0.014}$ | $0.065_{\pm 0.023}$ |
| Qwen3-14B | $0.007_{\pm 0.01}$ | $0.101_{\pm 0.016}$ | $0.27_{\pm 0.041}$ |
| Qwen3-30B-A3B | $0.108_{\pm 0.036}$ | $0.07_{\pm 0.014}$ | $0.232_{\pm 0.039}$ |
| Qwen3-32B | $0.038_{\pm 0.022}$ | $0.068_{\pm 0.014}$ | $0.105_{\pm 0.028}$ |
| Meta-Llama-3.1-8B-Instruct | $0.182_{\pm 0.045}$ | $0.025_{\pm 0.008}$ | $0.022_{\pm 0.014}$ |
| Mistral-Small-24B-Instruct-2501 | $0.122_{\pm 0.038}$ | $0.02_{\pm 0.008}$ | $0.038_{\pm 0.018}$ |

---

[6]Temperature of 0.0 will not generate fully deterministic result due to finite precision.

## C.2  PRM800K-SC RESULT IN 4,096 TOKEN BUDGET

To ensure a fair comparison between internal and external error correction, and across models, we maintain a fixed token budget of 1,024 across all conditions. This design choice partly isolates self-correction capabilities from the effect of test time compute, providing a more rigorous test of the blind spot phenomenon. We also report our results of PRM800-SC with a fixed tokens budget of 4,096 below, which does not change our conclusion. We do not report the result of SCLI5 and GSM8K-SC as the ratio of model responses exceeding 1,024 tokens is immaterial.

Table 14: Mean accuracy of models in PRM800K-SC at different compute budget

| Model | External Error | | Internal Error | | Appending "*Wait*" | |
| Compute budget | 1,024 | 4,096 | 1,024 | 4,096 | 1,024 | 4,096 |
|---|---|---|---|---|---|---|
| Llama-4-Maverick-17B-128E-Instruct-FP8 | 0.71 | 0.721 | 0.455 | 0.458 | 0.67 | 0.676 |
| DeepSeek-V3-0324 | 0.775 | 0.938 | 0.475 | 0.509 | 0.772 | 0.821 |
| Qwen2.5-72B-Instruct | 0.612 | 0.614 | 0.154 | 0.161 | 0.438 | 0.449 |
| Llama-4-Scout-17B-16E-Instruct | 0.58 | 0.578 | 0.263 | 0.257 | 0.545 | 0.542 |
| Llama-3.3-70B-Instruct | 0.359 | 0.366 | 0.246 | 0.257 | 0.46 | 0.469 |
| Qwen3-235B-A22B | 0.786 | 0.806 | 0.348 | 0.368 | 0.705 | 0.732 |
| phi-4 | 0.714 | 0.719 | 0.092 | 0.092 | 0.328 | 0.337 |
| Qwen2.5-7B-Instruct | 0.576 | 0.569 | 0.141 | 0.141 | 0.442 | 0.444 |
| Qwen2-7B-Instruct | 0.658 | 0.65 | 0.058 | 0.058 | 0.324 | 0.333 |
| Qwen3-14B | 0.705 | 0.743 | 0.254 | 0.268 | 0.696 | 0.746 |
| Qwen3-30B-A3B | 0.779 | 0.817 | 0.194 | 0.19 | 0.683 | 0.712 |
| Qwen3-32B | 0.754 | 0.781 | 0.083 | 0.085 | 0.527 | 0.522 |
| Meta-Llama-3.1-8B-Instruct | 0.181 | 0.183 | 0.02 | 0.02 | 0.194 | 0.203 |
| Mistral-Small-24B-Instruct-2501 | 0.496 | 0.498 | 0.016 | 0.016 | 0.27 | 0.277 |

## D PROMPT

### D.1 GENERATING GSM8K-SC

```python
from pydantic import BaseModel

class ReasoningWithMistake(BaseModel):
    reasoning_steps_with_one_mistake: List[str]
    mistake_step: int
    type_of_mistake: str
    description_of_mistake: str
    incorrect_answer: str
```

```
You are a helpful assistant that follow instructions. Output in
JSON format.
```

```
<question>
{question}
</question>

<reasoning_steps>
{reasoning_steps}
</reasoning_steps>

<answer>
{answer}
</answer>

<type_of_mistake>
{error_type}: {error_description}
</type_of_mistake>

You task is to introduce one mistake in step {mistake_step} in <
reasoning_steps> and arrive at an answer different from <answer>.
You will output:
- <reasoning_steps> with mistake
- the step that contains the mistake
- type of the mistake
- description of the mistake
- incorrect answer
```

Figure 11: Output schema, system prompt and prompt for generating GSM8K-SC dataset

```
from pydantic import BaseModel

class Calculation(BaseModel):
    incorrect_answer: float
```

```
You are a helpful assistant that follow instructions. Output in
JSON format.
```

```
<question>
{question}
</question>

<description_of_mistake>
{description_of_mistake}
</description_of_mistake>

<incorrect_reasoning>
{incorrect_reasoning}
</incorrect_reasoning>

You are given a question, a description of the mistake, and the
resulting incorrect reasoning.
Your task is to follow the incorrect reasoning to arrive at the
incorrect answer.
```

Figure 12: Output schema, system prompt and prompt for validating GSM8K-SC dataset

## D.2 AUTOMATIC EVALUATION

```
from pydantic import BaseModel

class Evaluation(BaseModel):
    is_correct_answer: bool
    is_correct_reasoning: bool
    is_backtracking: bool
    is_admitting_mistake: bool
```

```
You are a helpful assistant that follow instructions. Output in
JSON format.
```

Figure 13: Output schema and system prompt for automatic evaluation of error in model

```
# FOR SCLI5
<question>
{question}
</question>

<golden_answer>
{golden_answer}
</golden_answer>

<given_wrong_answer>
{given_wrong_answer}
</given_wrong_answer>

<completion_from_model>
{completion_from_model}
</completion_from_model>

The model was provided a wrong answer <given_wrong_answer>, and is
required to self correct and arrive at the golden answer.
You have to assess if <completion_from_model> :
- has backtracked
- has finally provided the answer that matches the <golden_answer>
- has provided correct reasoning
- has admitted the mistake.
```

```
# FOR GSM8K-SC AND PRM800K-SC
<question>
{question}
</question>

<golden_answer>
{golden_answer}
</golden_answer>

<given_wrong_reasoning>
{given_wrong_reasoning}
</given_wrong_reasoning>

<completion_from_model>
{completion_from_model}
</completion_from_model>

The model was provided a wrong step in the reasoning process <
given_wrong_reasoning>, and is required to self correct and arrive
at the golden answer.
You have to assess if <completion_from_model> :
- has backtracked
- has finally provided the answer that matches the <golden_answer>
- has provided correct reasoning
- has admitted the mistake.
```

Figure 14: Prompt for automatic evaluation of error in model

```
from pydantic import BaseModel

class EvaluationErrorInUser(BaseModel):
    is_correct_answer: bool
    is_correct_critique: bool
```

```
You are a helpful assistant that follow instructions. Output in
JSON format.
```

Figure 15: Output schema and system prompt for automatic evaluation of error in user

```
# FOR SCLI5
<question_and_user_answer>
{question_and_user_answer}
</question_and_user_answer>

<golden_answer>
{golden_answer}
</golden_answer>

<response_from_model>
{response_from_model}
</response_from_model>

The model was provided with <question_and_user_answer> from user.
You have to assess if <response_from_model> :
- contains correct answer that matches the <golden_answer>
- contains correct critique of <question_and_user_answer>
```

```
# FOR GSM8K-SC AND PRM800K-SC
<question_and_user_reasoning>
{question_and_user_reasoning}
</question_and_user_reasoning>

<golden_answer>
{golden_answer}
</golden_answer>

<response_from_model>
{response_from_model}
</response_from_model>

The model was provided with <question_and_user_reasoning> from user
.
You have to assess if <response_from_model> :
- contains correct answer that matches the <golden_answer>
- contains correct critique of <question_and_user_reasoning>
```

Figure 16: Prompt for automatic evaluation of error in user

# E EXAMPLE OF SELF-CORRECTION BENCH

## E.1 SCLI5

```
# error in user prompt
[
    {
        "content": "What is the answer of 20 + 1? The answer is 22.",
        "role": "user"
    }
]

# error in model response
[
    {
        "content": "What is the answer of 20 + 1?",
        "role": "user"
    },
    {
        "content": "The answer is 22.",
        "role": "assistant"
    }
]
```

## E.2 GSM8K-SC

```
# error in user prompt
[
    {
        "content": "Harry slept 9 hours last night. His friend James slept only 2/3
        of what Harry slept. How many more hours did Harry sleep than James? James
        slept 9 hours / 2/3 = 13.5 hours. So, Harry slept 9 - 13.5 = -4.5 hours more
         than James.",
        "role": "user"
    }
]

# error in model response
[
    {
        "content": "Harry slept 9 hours last night. His friend James slept only 2/3
        of what Harry slept. How many more hours did Harry sleep than James?",
        "role": "user"
    },
    {
        "content": "James slept 9 hours / 2/3 = 13.5 hours. So, Harry slept 9 - 13.5
         = -4.5 hours more than James.",
        "role": "assistant"
    }
]
```

### E.3  PRM800K-SC

```
# error in user prompt
[
   {
      "content": "A Senate committee has 5 Democrats, 5 Republicans, and 1
      Independent. In how many ways can they sit around a circular table if all
      the members of each party all sit next to each other? (Two seatings are
      considered equivalent if one is a rotation of the other.) I notice that
      there are three groups of people: Democrats, Republicans, and Independent. I
       wonder how many ways I can arrange these groups around the circle, ignoring
       the order within each group for now. I recall that the number of ways to
      arrange n distinct objects around a circle is (n-1)!, since we can fix one
      object and then permute the rest. So, for the three groups, there are (3-1)!
       = 2! = 2 ways to arrange them around the circle. For example, one way is D-
      R-I, and the other way is R-D-I, where D stands for Democrats, R for
      Republicans, and I for Independent. Now, I need to consider the order within
       each group. For the Democrats, there are 5! ways to order them in a line,
      but only half of them are distinct around a circle, since reversing the
      order gives the same seating. So, there are 5!/2 = 60 ways to order the
      Democrats around a circle. Similarly, there are 60 ways to order the
      Republicans around a circle. For the Independent, there is only one way to
      order him or her around a circle. So, to get the total number of ways to
      seat the committee, I need to multiply the number of ways to arrange the
      groups by the number of ways to order each group. That gives me 2 x 60 x 60
      x 1 = 7200 ways.",
      "role": "user"
   }
]
# error in model response
[
   {
      "content": "A Senate committee has 5 Democrats, 5 Republicans, and 1
      Independent. In how many ways can they sit around a circular table if all
      the members of each party all sit next to each other? (Two seatings are
      considered equivalent if one is a rotation of the other.)",
      "role": "user"
   },
   {
      "content": "I notice that there are three groups of people: Democrats,
      Republicans, and Independent. I wonder how many ways I can arrange these
      groups around the circle, ignoring the order within each group for now. I
      recall that the number of ways to arrange n distinct objects around a circle
       is (n-1)!, since we can fix one object and then permute the rest. So, for
      the three groups, there are (3-1)! = 2! = 2 ways to arrange them around the
      circle. For example, one way is D-R-I, and the other way is R-D-I, where D
      stands for Democrats, R for Republicans, and I for Independent. Now, I need
      to consider the order within each group. For the Democrats, there are 5!
      ways to order them in a line, but only half of them are distinct around a
      circle, since reversing the order gives the same seating. So, there are 5!/2
       = 60 ways to order the Democrats around a circle. Similarly, there are 60
      ways to order the Republicans around a circle. For the Independent, there is
       only one way to order him or her around a circle. So, to get the total
      number of ways to seat the committee, I need to multiply the number of ways
      to arrange the groups by the number of ways to order each group. That gives
      me 2 x 60 x 60 x 1 = 7200 ways.",
      "role": "assistant"
   }
]
```

