# OpenReview forum: "Self-Correction Bench: Uncovering and Addressing the Self-Correction Blind Spot in Large Language Models"
_ICLR.cc/2026/Conference — Submitted to ICLR 2026_

### Official Review · Reviewer_DyNA · 2025-10-25

**Soundness:** 3
**Presentation:** 3
**Contribution:** 2
**Rating:** 4
**Confidence:** 4

**Summary:**

The authors target a known limitation in LLMs, self-correction, wherein it is harder to find errors in their own outputs, compared to finding similar / identical errors from external sources. They term this limitation Self-Correction Blind Spot, and introduce Self-Correction Bench. They argue that since human demonstrations rarely include self-correction, models only learn error-correction through outcome feedback during the RL phase. They also mention that self-correction is dormant in language models, and appending “Wait” improves the models ability to overcome blind spots.

**Strengths:**

- Studying self reflection is a very relevant problem for LLMs.
- The experiment of injecting incorrect responses in the output of the model and removing the stop token, and comparing it with the injection in the prompt is interesting.
- The results in Appendix C, indicating that the results don’t change with temperature is intriguing.
- Reducing the possibility for knowledge gaps by using easy, medium, and hard datasets is a good progressive difficulty metric. This isolates the self-correction capability from confounding factors.
- The author’s conclusion that reasoning models are better than non reasoning models on self correction is intuitive. Also the fact that correction markers can reduce the gap.
- Self-Correction Bench can be a useful benchmarks for identifying reflection capabilities in models.

**Weaknesses:**

- Even though the conclusions about corrective markers is interesting, the paper does not introduce any theoretical / technical contributions.
- The benchmarks are very limited, to SCLI5, GSM8K and MATH. It would be interesting to see this across different tasks like code, logic, and multimodal reasoning, etc.
- Overall, while this dataset has potential to be a useful benchmark for studying LLM reflection capabilities, the paper lacks strong technical backing and explanations, and is also very limited in the domains it is targeting.

**Questions:**

- It is unclear how authors are measuring Self Correction scores before and after committing to an answer. Is it using an LLM as a judge? More explanation is required here.

---

> ### Author Response · Authors · 2025-11-21
> **Response to Reviewer DyNA**
>
> We sincerely thank Reviewer DyNA for recognizing the experimental design quality and the work's relevance. We greatly appreciate the recognition of this dataset’s potential to be a useful benchmark.  We address your concerns below with additional experiments and clarifications.
>
> >W1: the paper does not introduce any theoretical / technical contribution
>
> We have addressed the general response with additional experiments. To conclude, our paper provides systematic evidence for understanding why the self-correction blind spot exists: we demonstrate strong correlations between the absence of error-correction sequences in training data and blind spot magnitude. Critically, we provide causal evidence that fine-tuning with error and self-correction data substantially reduces the blind spot. This leads to practical training recipe change - to include error and self-correction data during model training, echoing our insight through the marginalisation of error states.
>
> > W2: It would be interesting to see this across different tasks like code, logic, and multimodal reasoning, etc
>
> We agree that expanding beyond mathematical reasoning would strengthen our work. We provide both theoretical arguments and empirical validation:
>
> **Theoretical argument**: The self-correction blind spot from training data composition. SFT datasets undergo quality filtering and remove errors across all domains (code, text, math). Error-correction sequences are absent regardless of task type. Therefore, error-free training data is a universal property of SFT datasets, not specific to mathematical reasoning.  This domain-agnostic mechanism (absence of error-correction in training leading to activation failure) predicts the blind spot should appear wherever SFT data lacks error sequences.
>
> **Empirical validation**: [Additional experiments] We empirically validate this prediction. We leverage BIG-Bench Mistake[1], generated using PaLM 2, that contains mistake in reasoning trace. We selected Tracking Shuffled Objects and Logical Deduction, which are unrelated to mathematical reasoning. We tested models from diverse model families (Llama3.1, Qwen3, Mistral-Small, Llama3.3 and DeepSeek-R1-Distill-Llama).
>
> ### Tracking shuffled subject
> | model                                     |   error_in_model |   error_in_user |   Blind Spot |   Size |
> |:------------------------------------------|-----------------:|----------------:|-------------:|-------:|
> | Meta-Llama-3.1-8B-Instruct     |           0.0731 |          0.2462 |       0.7031 |    260 |
> | Qwen3-14B                            |           0.0962 |          0.9962 |       0.9035 |    260 |
> | Mistral-Small-24B-Instruct-2501 |           0.2    |          0.5538 |       0.6389 |    260 |
> | Llama-3.3-70B-Instruct         |           0.1269 |          0.9308 |       0.8636 |    260 |
> | DeepSeek-R1-Distill-Llama-70B |           0.95   |          0.9731 |       0.0237 |    260 |
>
> ### Logical deduction
> | model                                     |   error_in_model |   error_in_user |   Blind Spot |   Size |
> |:------------------------------------------|-----------------:|----------------:|-------------:|-------:|
> | Meta-Llama-3.1-8B-Instruct     |           0.1395 |          0.2075 |       0.3279 |    294 |
> | Qwen3-14B                            |           0.2619 |          0.8537 |       0.6932 |    294 |
> | Mistral-Small-24B-Instruct-2501 |           0.2551 |          0.449  |       0.4318 |    294 |
> | Llama-3.3-70B-Instruct         |           0.3707 |          0.6939 |       0.4657 |    294 |
> | DeepSeek-R1-Distill-Llama-70B |           0.6939 |          0.9048 |       0.2331 |    294 |
>
> Our experiment confirms that the blind spot exists across domains. Notably we observe model finetuned with self-correction data drastically reduces self-correction blind spot when we compare DeepSeek-R1-Distill-Llama-70B vs its base model Llama-3.3-70B-Instruct.
>
>
> **Action**: We will include these experiment results in the revised manuscript.
>
> >Q1: "It is unclear how authors are measuring Self Correction scores before and after committing to an answer. Is it using an LLM as a judge?"
>
> Yes, we use an LLM as judge for automatic evaluation. We provide two versions of error injection—without and with committing to the wrong answer, prompting tested models separately. The LLM is instructed to compare answers against ground truth, a straightforward and objective task. We verified evaluation quality through manual review of 100 samples per dataset (Section 5.1), confirming minimal bias.
>
> We greatly appreciate your thoughtful review. We hope this demonstrates that our work provides substantial theoretical contributions beyond empirical observation, and our additional experiment to extend beyond mathematical reasoning has addressed your concern.
>
> [1] G Tyen. LLMs cannot find reasoning errors, but can correct them given the error location. Findings of the Association for Computational Linguistics: ACL.

---

### Official Review · Reviewer_ZE7F · 2025-10-28

**Soundness:** 3
**Presentation:** 3
**Contribution:** 2
**Rating:** 4
**Confidence:** 4

**Summary:**

The paper studies the self-correction capability of large-language models (LLMs) by: 1) creating a dataset of self-correction prompts, 2) showing empirically that some LLMs do poorly on self-correction prompts (referred to as a self-correction "blind spot" in the paper), and 3) showing empirically that appending the self-correction prompts with certain conditioning tokens (such as the word "wait") can improve the self-correction performance of models that did poorly initially. A variety of non-reasoning and reasoning models were studied, across a number of model families (such as Qwen, Gemma, and DeepSeek). The self-correction prompts are all based on math problems, ranging from the trivial (SCLI5) to quite difficult.

**Strengths:**

1. Dataset contribution specifically focused on self-correction (or "correction", see weakness 1) by asking models to determine whether the given answer or reasoning in the prompt needs to be corrected, and indeed make the correction. This allows for the analysis in the paper to examine the correction ability of both reasoning and non-reasoning LLMs.

2. Empirical comparative analysis of the correction capability of reasoning and non-reasoning LLMs reveals that one strong differentiating factor that specifically makes reasoning models better in performance is their ability to recognize when an error is present (in a given answer or reasoning trace), and make the necessary correction. Whereas, as this paper shows, non-reasoning models cannot.

3. Empirical evidence for inserting conditioning tokens such as "Wait" that alerts a non-reasoning model to the fact that the given answer may not be correct can improve their correction performance, making up the gap to or even exceeding the performance of reasoning models. It is quite clear that even non-reasoning models can give the correct answer when they are made aware that correction is required.

**Weaknesses:**

1. The dataset does not strictly study self-correction. By the construction description, all 3 sub-datasets were generated by inserting wrong answers or reasoning traces into a given prompt from standard datasets (with possibly a short sequence of model output) using off-the-self closed-source models (such as GPT 4.1). Since the incorrect answers were not generated by the models-under-test (open-source models with fewer parameters), it is unknown how likely they are under each model's own sampling distributions, this is not self-correction (i.e., the model may not even sample the incorrect solution). A proper self-correction dataset would sample strictly on-policy from each model and filter for incorrect answers.

The datasets are still valuable as benchmarks for correcting an off-policy incorrect answer or reasoning trace, but they do not study "self" correction.

2. No actionable conclusions for reasoning models. The main result and contribution of the paper is that non-reasoning models have a self-correction blindspot, which could be corrected by conditioning on certain tokens. Moreover, reasoning models have small or no self-correction blindspot. As reasoning models are the stronger of these two types at solving math problems, this weakens the significance of the key findings of the work (i.e., Why not just use reasoning models?).

**Questions:**

Please show more examples of the self-correction dataset, e.g., 1 for each subset, and for when there are reasoning steps.

---

> ### Author Response · Authors · 2025-11-21
> **Response to Reviewer ZE7F**
>
> We sincerely thank Reviewer ZE7F for the thoughtful review. We greatly appreciate your recognition of the work's potential to be a valuable benchmark and address your concerns below.
>
> > W1: A proper self-correction dataset would sample strictly on-policy from each model and filter for incorrect answers.
>
> We have addressed the reviewer’s concern in the general comment. We acknowledge the limitation of using off-policy error, but we conclude its benefit outweighs the cons. Moreover, we tested 3 models (Qwen2-7B-Instruct, Qwen2.5-7B-Instruct, Llama-3.1-8B-Instruct) on their own on-policy errors from ProcessBench. When these on-policy errors are presented externally, models achieve 14-59% correction accuracy, demonstrating underlying correction capability. Our framework predicts that when these same errors are presented internally, the blind spot substantially reduces this correction rate, which our main experiments confirm. This validates that the phenomenon exists for on-policy errors.
>
> **Action**: We will include these additional results in the revised manuscript.
>
> > W2: There are no really actionable conclusions about reasoning models since they are basically not subject to this problem... So why not just use reasoning models?
>
> This is an excellent practical question. We address it from multiple perspectives:
>
> 1. Role of non-reasoning models
>
> Non-reasoning models still matter because of cost and latency concerns as reasoning models require significant compute, making it impractical for many production scenarios. Our benchmark suggests non-reasoning models can benefit substantially from including error correction data in training.
>
> 2. Actionable insight for reasoning models
>
> a) Understanding the mechanism: Our analysis reveals that reasoning models' advantage stems from training with error-correction sequences (99% contain ≥1 correction marker vs. <1% in standard SFT). This provides a deeper understanding of why RL-trained models perform better as they learn to generate correction markers that activate dormant self-correction capabilities already present in base models.
>
> b) Efficiency evaluation: Even reasoning models show imperfect performance (far from 91%+ accuracy on our benchmark). Our error-injection methodology enables targeted evaluation of:
> - Error correction in specific domains
> - Self-correction efficiency (tokens required for self-correction)
>
> c) Architecture and training insights: Our comparison of Qwen3's thinking vs. non-thinking modes shows that despite continuous fine-tuning to fuse both modes, non-thinking mode retains the blind spot. This suggests architectural and training approaches deserve further investigation.
>
>
> We sincerely thank the reviewer for these thoughtful questions.  We hope our additional on-policy validation experiments and actionable insights for both model types address your concerns and demonstrate the broader value of our work.
>
> We will provide the examples of the Self-correction Bench in a subsequent response.

---

> > ### Author Response · Authors · 2025-11-21
> > **Examples of the self-correction dataset**
> >
> > >Q1: Please show more examples of the self-correction dataset
> >
> > ## SCLI5
> > ```
> > # error in user prompt
> > [
> > {
> > "content": "What is the answer of 20 + 1? The answer is 22.",
> > "role": "user"
> > }
> > ]
> >
> > # error in model response
> > [
> > {
> > "content": "What is the answer of 20 + 1?",
> > "role": "user"
> > },
> > {
> > "content": "The answer is 22.",
> > "role": "assistant"
> > }
> > ]
> > ```
> > ## GSM8K_SC
> > ```
> > # error in user prompt
> > [
> > {
> > "content": "Harry slept 9 hours last night. His friend James slept only 2/3 of what Harry slept. How many more hours did Harry sleep than James? James slept 9 hours / 2/3 = 13.5 hours. So, Harry slept 9 - 13.5 = -4.5 hours more than James.",
> > "role": "user"
> > }
> > ]
> >
> > # error in model response
> > [
> > {
> > "content": "Harry slept 9 hours last night. His friend James slept only 2/3 of what Harry slept. How many more hours did Harry sleep than James?",
> > "role": "user"
> > },
> > {
> > "content": "James slept 9 hours / 2/3 = 13.5 hours. So, Harry slept 9 - 13.5 = -4.5 hours more than James.",
> > "role": "assistant"
> > }
> > ]
> > ```
> >
> > ## PRM800K_SC
> > ```
> > # error in user prompt
> > [
> > {
> > "content": "A Senate committee has 5 Democrats, 5 Republicans, and 1 Independent. In how many ways can they sit around a circular table if all the members of each party all sit next to each other? (Two seatings are considered equivalent if one is a rotation of the other.) I notice that there are three groups of people: Democrats, Republicans, and Independent. I wonder how many ways I can arrange these groups around the circle, ignoring the order within each group for now. I recall that the number of ways to arrange n distinct objects around a circle is (n-1)!, since we can fix one object and then permute the rest. So, for the three groups, there are (3-1)! = 2! = 2 ways to arrange them around the circle. For example, one way is D-R-I, and the other way is R-D-I, where D stands for Democrats, R for Republicans, and I for Independent. Now, I need to consider the order within each group. For the Democrats, there are 5! ways to order them in a line, but only half of them are distinct around a circle, since reversing the order gives the same seating. So, there are 5!/2 = 60 ways to order the Democrats around a circle. Similarly, there are 60 ways to order the Republicans around a circle. For the Independent, there is only one way to order him or her around a circle. So, to get the total number of ways to seat the committee, I need to multiply the number of ways to arrange the groups by the number of ways to order each group. That gives me 2 x 60 x 60 x 1 = 7200 ways.",
> > "role": "user"
> > }
> > ]
> >
> > # error in model response
> > [
> > {
> > "content": "A Senate committee has 5 Democrats, 5 Republicans, and 1 Independent. In how many ways can they sit around a circular table if all the members of each party all sit next to each other? (Two seatings are considered equivalent if one is a rotation of the other.)",
> > "role": "user"
> > },
> > {
> > "content": "I notice that there are three groups of people: Democrats, Republicans, and Independent. I wonder how many ways I can arrange these groups around the circle, ignoring the order within each group for now. I recall that the number of ways to arrange n distinct objects around a circle is (n-1)!, since we can fix one object and then permute the rest. So, for the three groups, there are (3-1)! = 2! = 2 ways to arrange them around the circle. For example, one way is D-R-I, and the other way is R-D-I, where D stands for Democrats, R for Republicans, and I for Independent. Now, I need to consider the order within each group. For the Democrats, there are 5! ways to order them in a line, but only half of them are distinct around a circle, since reversing the order gives the same seating. So, there are 5!/2 = 60 ways to order the Democrats around a circle. Similarly, there are 60 ways to order the Republicans around a circle. For the Independent, there is only one way to order him or her around a circle. So, to get the total number of ways to seat the committee, I need to multiply the number of ways to arrange the groups by the number of ways to order each group. That gives me 2 x 60 x 60 x 1 = 7200 ways.",
> > "role": "assistant"
> > }
> > ]
> > ```
> >
> > **Action**: We will include 1 example from each subset of our benchmark in the revised manuscript.

---

> > ### Author Response · Authors · 2025-11-27
> > **Update to On-policy Error Validation Additional Experiment**
> >
> > **Correction Notice**
> > We have posted an update to our general response regarding the on-policy error validation experiment.
> >
> > **Summary**:
> > - Issue: Implementation error in prompt format (truncated vs. full traces)
> > - Impact: Corrected rates are 6-10% (previously reported 14-59%)
> > - Scope: Affects only supplementary rebuttal analysis, not part of existing manuscript.
> > - Core finding: Unchanged - blind spots exist for on-policy errors
> >
> > The complete result and interpretation are available in our updated general response.
> >
> > We appreciate Reviewer ZE7F's understanding.

---

### Official Review · Reviewer_57Bc · 2025-11-01

**Soundness:** 1
**Presentation:** 2
**Contribution:** 2
**Rating:** 2
**Confidence:** 4

**Summary:**

This paper studies LLMs' self-correction blind spot and introduces Self-Correction Bench, an evaluation framework to measure this phenomenon through controlled error injection.

**Strengths:**

1. For originality and significance, I think the newly introduced evaluation framework is a nice addition to existing works.

2. The paper is generally understandable, but I need more explanation/analysis described in the weaknesses and questions sections below.

**Weaknesses:**

1. The paper discusses self-correction blind spot on LLMs, but closed-source LLMs are unfortunately not studied. Although it is explained as "close-source models lack support for fine-grained control of prefix inject critical for our methodology" in line 236, I do not think this would stop you from studying closed-source models. You may want to analyze their reasoning chains directly and compare model outputs side-by-side. Otherwise, the findings of this paper is very limited.

2. The analysis offered appears to be coarse and is not very comprehensive. You mentioned that "this limitation may be influenced by training data", but I feel there can be much more reasons. For example, an LLM may tend to have relatively lower uncertainty on its own tokens than on tokens generated by another model.

3. I do not see any quality check on the benchmark introduced by the authors in Section 4, so I am not very confident with its overall quality. A common practice of similar papers is to leverage human evaluation to do some manual verification.

4.  A “Wait” prompt is not a novel idea. For example, [1] explores an ensemble of critics and the model's own feedback. This is a 2-year-old work. You may find more papers talking about something similar to your prompt.

[1] "N-Critics: Self-Refinement of Large Language Models with Ensemble of Critics"

**Questions:**

1. Is it possible to design some customized methods to analyze closed-source LLMs?

2. Do you think there are any other reasons of self-correction blind spot besides training data?

---

> ### Author Response · Authors · 2025-11-21
> **Response to Reviewer 57Bc**
>
> We thank Reviewer 57Bc for the constructive feedbacks. We address each of your concerns below with additional experiments.
> > W1: but closed-source LLMs are unfortunately not studied
>
> We appreciate this concern and have made efforts to address it within the constraints of available APIs.
> Most of the frontier LLM does not officially support assistant response prefill because it heightens security risk, except Anthropic which explicitly supports prefilling claude response.
>
> [Additional experiment] As such, we successfully evaluated Sonnet 4 and Haiku.
> | model                     | dataset    |   error_in_model |   error_in_user |   Blind Spot |
> |:--------------------------|:-----------|-----------------:|----------------:|-------------:|
> | claude-3-5-haiku-20241022 | scli5      |           0.6923 |          0.8217 |       0.1574 |
> |  | gsm8k_sc   |           0.3283 |          0.8842 |       0.6288 |
> |  | prm800k_sc |           0.0804 |          0.3817 |       0.7895 |
> | claude-sonnet-4-20250514  | scli5      |           0.4266 |          0.9965 |       0.5719 |
> |   | gsm8k_sc   |           0.655  |          0.9741 |       0.3276 |
> |   | prm800k_sc |           0.5759 |          0.8772 |       0.3435 |
>
> Both models exhibit substantial self-correction blind spot, strongly suggesting this phenomenon generalises to frontier models.
>
> **Action**: We will include the result in our revised manuscript.
>
> > W2: The analysis offered appears to be coarse and is not very comprehensive. You mentioned that "this limitation may be influenced by training data", but I feel there can be much more reasons. For example, an LLM may tend to have relatively lower uncertainty on its own tokens than on tokens generated by another model.
>
> We have addressed the reviewer’s concern in the general comment in depth. Specifically, controlled experiments with DeepSeek-R1-Distill models demonstrate that fine-tuning with error-correction data reduces blind spots dramatically, providing causal evidence.
>
> Regarding the token-uncertainty hypotheses (see general response section "On-policy error is not used"), we additionally tested 3 models (Qwen2-7B-Instruct, Qwen2.5-7B-Instruct, Llama-3.1-8B-Instruct) on their own on-policy errors from ProcessBench. We empirically observe the self-correction blind spot exists even for errors the model itself generated, achieving 14-59% correction accuracy when presented externally but substantially lower when presented internally. This suggests uncertainty differences alone cannot explain the phenomenon.
>
> > W3: I do not see any quality check on the benchmark introduced by the authors in Section 4
>
> We agree that robust quality validation is paramount.
> - SCLI5: programmatically generated → quality guaranteed by construction
> - GSM8K-SC: we built on peer-reviewed GSM8K. We use the ground truth to measure accuracy when error is injected. For the error generation, the objective is to make sure the error exists and the error itself is consistent for realism. We use ‘gpt4.1-2025-04-14’ to generate errors and ‘gemini-2.5-flash-preview-05- 20 to validate that by following the given incorrect reasoning, one could lead to the inconsistent answers. 6 mismatches were found and therefore dropped. We programmatically verify that the incorrect answer is different from ground truth.
> - For PRM800K-SC: sampled from peer-reviewed PRM800K.
>
> Crucially, we released the codes, enabling full reproducibility and auditing. Furthermore, the fact that our benchmark successfully shows highly correlated blind spot across three levels confirms its internal consistency and quality as a diagnostic tool.
>
> > W4: A 'Wait' prompt is not a novel idea. For example, [N-Critics] explores ensemble of critics…
>
> We appreciate this context and clarify our contribution relative to prior work:
> N-Critics (2023) approach involves multiple critic models in ensemble architecture, and involves multiple inference passes; in contrast, our approach focuses on a single inference pass from a single model with a single word intervention ("Wait", "But", "However").
>
> Our contributions:
> - Isolating and quantifying a 64.5% average blind spot that is activation-dependent, not knowledge-dependent.
> - Showing a one-word trigger activates dormant correction capability.
> - Providing causal evidence (DeepSeek-R1-Distill experiments) that the fix is simply including error-correction trajectories in SFT — a concrete training recipe.
>
> **Action**: We will include N-Critics and related work in our revised manuscript with appropriate context.
>
> We sincerely thank the reviewer for review that helped us improve our work. We hope that our response with additional experiments on close-sourced frontier model, more in-depth analysis could address most of the concerns and also to clarify our contribution to the community.

---

> > ### Author Response · Authors · 2025-11-27
> > **Update to On-policy Error Validation Additional Experiment**
> >
> > **Correction Notice**
> > We have posted an update to our general response regarding the on-policy error validation experiment.
> >
> > **Summary**:
> > - Issue: Implementation error in prompt format (truncated vs. full traces)
> > - Impact: Corrected rates are 6-10% (previously reported 14-59%)
> > - Scope: Affects only supplementary rebuttal analysis, not part of existing manuscript.
> > - Core finding: Unchanged - blind spots exist for on-policy errors
> >
> > The complete result and interpretation are available in our updated general response.
> >
> > We appreciate Reviewer 57Bc's understanding.

---

### Official Review · Reviewer_EoiX · 2025-11-02

**Soundness:** 3
**Presentation:** 3
**Contribution:** 2
**Rating:** 2
**Confidence:** 3

**Summary:**

The paper focuses on isolating a failure in modern LLMs called "Self-correction Blind Spot", where the model can correct an error if attributed to an external factor (e.g. a user or a tool), but fails to do so for itself. It does so by quantifying the phenomena on 3 benchmarks of increasing complexity (SCLI5, GSM8K-SC, and PRM800K-SC).

It then points out that the lack of correction is due to a lack of activation (or intent to fix), rather than the inability to fix (e.g. not having the knowledge). Minimal intervention is design succesfully, namely by using the word "Wait."

**Strengths:**

There are several strengths to the manuscript:
1. Rather clear isolation of the problem and extensive empirical analysis of multiple models on the behavior. The experiment design seems reasonable, where exact same sequence of tokens is presented but with different attribution to observe differing model behavior.
2. The introduction of a toy benchmark (SCLI5) to explain clearly the phenomena, followed by studying on real-world benchmarks.
3. Identifying an intervention, e.g. the word "wait".

**Weaknesses:**

I believe the overall contribution is somewhat naive and does not go sufficiently in-depth in understanding the mechanics of the model behavior. The weaknesses I would like to hear the author's opinion are:

1. The entire work is prompt engineering - from the detection to the solution. The authors acknowledge this, but I find it necessary to go a step further and point out some training recipe changes that mitigates this to some degree. Naive fine-tuning with the word 'wait' may not work, while asking all researchers / developers to start using 'wait' seems also non-ideal.

2. An off-policy (in the RL sense) / biased setup by design. The generation of the high-quality incorrect reasoning traces by another model and then feeding them into a current model making them looking like it's own output is somewhat flawed by design. The generated tokens would have to be from the model itself (e.g. it's weights) or otherwise the sequence would always be off-policy of the model state (and hence can't be used for claiming the blind spot).

**Questions:**

Please refer to weaknesses.

Furthermore, can the code be released? Unfortunately the repository at 4openscience is returning an error for all files, despite some directory structure existing.

---

> ### Author Response · Authors · 2025-11-21
> **Response to Reviewer EoiX**
>
> We sincerely thank Reviewer EoiX for the thoughtful assessment and positive ratings on soundness (3: good) and presentation (3: good). We have conducted substantial additional experiments to address your concerns about depth and methodology.
> > W1: I believe the overall contribution is somewhat naive and does not go sufficiently in-depth in understanding the mechanics... The entire work is prompt engineering - from detection to solution.
>
> We have addressed the reviewer’s concern in the general comment in depth. Specifically, controlled experiments with DeepSeek-R1-Distill models demonstrate that fine-tuning with error-correction data reduces blind spots dramatically, providing causal evidence for a training-level solution beyond prompt engineering.
> > W2: The generation of the high-quality incorrect reasoning traces by another model and then feeding them into a current model making them looking like it's own output is somewhat flawed by design
>
> We have addressed the reviewer’s concern in the general comment. We acknowledge the limitation of using off-policy error, but we conclude its benefit outweighs the cons. Moreover, we tested 3 models (Qwen2-7B-Instruct, Qwen2.5-7B-Instruct, Llama-3.1-8B-Instruct) on their own on-policy errors from ProcessBench. When these on-policy errors are presented externally, models achieve 14-59% correction accuracy, demonstrating underlying correction capability. Our framework predicts that when these same errors are presented internally, the blind spot substantially reduces this correction rate, which our main experiments confirm. This validates that the phenomenon exists for on-policy errors.
> > Q1: Furthermore, can the code be released? Unfortunately the repository at 4openscience is returning an error for all files, despite some directory structure existing.
>
> We verified the [link](https://anonymous.4open.science/r/self-correction-bench-1B58/README.md) is functional.
> We have also provided an alternative via [google drive](https://drive.google.com/file/d/1BPdJ9o1__oCXDXWy0Audn4gGoK0uYaL3/view?usp=sharing). Please let us know if the problem persists.
>
> We sincerely thank Reviewer EoiX for the thorough and thoughtful assessment that helps strengthen our work. We hope that our response with additional experiments addresses most of the concerns and also to clarify our contribution to the community. We commit to update the manuscript with a greater emphasis on systematic understanding and discussion of on-policy and off-policy, backed by the additional experiment we have conducted.

---

> > ### Author Response · Authors · 2025-11-27
> > **Update to On-policy Error Validation Additional Experiment**
> >
> > **Correction Notice**
> > We have posted an update to our general response regarding the on-policy error validation experiment.
> >
> > **Summary**:
> > - Issue: Implementation error in prompt format (truncated vs. full traces)
> > - Impact: Corrected rates are 6-10% (previously reported 14-59%)
> > - Scope: Affects only supplementary rebuttal analysis, not part of existing manuscript.
> > - Core finding: Unchanged - blind spots exist for on-policy errors
> >
> > The complete result and interpretation are available in our updated general response.
> >
> > We appreciate Reviewer EoiX's understanding.

---

### Author Response · Authors · 2025-11-21
**General Response to All Reviewers (1/2)**

We sincerely thank the reviewers for their precious time and constructive feedback. We are very encouraged by the feedback that our work studies a very relevant area in LLM today and recognition of Self-correction Bench’s potential to be a useful benchmark that could benefit researchers and practitioners.

Across different reviews, we identified two common themes (1) Lack of depth and contribution (2) on-policy error is not used, which are addressed below.

## (1) Lack of Depth of Analysis and Contribution (EoiX, 57Bc, DyNA)
### Systematic Understanding: Errors are Absent from SFT Data
In section 3.1, we start our framework with “Error-free generation is a **special** case of this framework - not the only path to correctness” through the marginalization of error states, which leads to the training recipe insight: including error and self-correction data as stated in section 7.

We note that SFT models trained mostly on error-free demonstration due to human preference. We demonstrate this in section 6.3 where correction markers are very limited in SFT dataset. In particular, the 90th percentile of correction marker frequency of OpenAssistant (both demonstrated and rated by human) is 1. In stark contrast, at least 99% of Mixture-of-Thoughts (distilled from DeepSeek-R1) and OpenThought3-1.2M (distilled from QwQ-32B) contains at least 1 correction marker.

Although popular open source model Llama3.1 and Qwen2.5 does not open source their SFT data, from their technical report, Llama3.1 considers samples that obtain maximum score as rated by Llama 3 as high quality while Qwen3 only retains responses deemed flawless by all scoring systems. It suggests responses with error and self-correction are filtered away.

### Causal Intervention
Inspired by the presence of correction markers when error is present in user prompt, we conducted a causal intervention study in section 6.1 to show that by appending “Wait”, “But” and “However” can mitigate self-correction blind spot. So naturally if we can make LLM predict correction markers as the next token upon seeing the error, we could activate self-correction. We can achieve so by finetuning LLM with error and self-correction data, such as data generated by reasoning models.

### Can SFT with error and self-correction data mitigate the blind spot?
[Additional experiment] We **additionally** evaluate DeepSeek-R1-Distill-Llama-8B and DeepSeek-R1-Distill-Llama-70B. Both models are fine-tuned respectively from Llama-3.1-8B and Llama-3.3-70B-Instruct with 800k SFT data (600k reasoning trajectories generated by DeepSeek-R1 and 200k training samples unrelated to reasoning) for 2 epochs. While we cannot isolate error-correction data as the sole causal factor (as the training includes other components), the dominant intervention is the addition of error-correction sequences. The results show dramatic blind spot reductions, providing strong causal evidence that fine-tuning with error and self-correction data substantially mitigates this limitation.

| Model                                              | Comparison Model |  Dataset   |   Accuracy Internal Error |  Accuracy External Error |   Blind Spot |   Blind Spot (Comparison Model) |
|:---------------------------------------------------|:----------|:----------|-----------------:|----------------:|-------------:|--------------------------:|
| DeepSeek-R1-Distill-Llama-8B  | Llama-3.1-8B-Instruct*| scli5     |           0.4615 |          0.9056 |       0.4903 |                    0.8289 |
| | | gsm8k     |           0.5994 |          0.6923 |       0.1342 |                    0.9708 |
| | | prm800k   |           0.4888 |          0.4911 |       0.0045 |                    0.8889 |
| DeepSeek-R1-Distill-Llama-70B | Llama-3.3-70B-Instruct | scli5     |           0.8497 |          0.958  |       0.1131 |                    0.4577 |
| || gsm8k     |           0.9162 |          0.8888 |      -0.0308 |                    0.6885 |
| | | prm800k   |           0.6562 |          0.625  |      -0.05   |                    0.3168 |
*Llama-3.1-8B-Instruct and DeepSeek-R1-Distill-Llama-8B share the same base model Llama-3.1-8B

We note the blind spot in SCLI5 is larger than GSM8k_SC and PRM800K_SC, which may be due to lower realism of error injected.

To conclude, our paper provides systematic evidence for understanding why the self-correction blind spot exists: we demonstrate strong correlations between the absence of error-correction sequences in training data and blind spot magnitude. Critically, we provide causal evidence that fine-tuning with error and self-correction data substantially reduces the blind spot. This leads to practical training recipe change - to include error and self-correction data during model training, echoing our insight through the marginalisation of error states.

**Action**: We commit to emphasize our contribution and append the result of additional experiments in our manuscript.

(to continue...)

---

> ### Author Response · Authors · 2025-11-21
> **General Response to All Reviewers (2/2)**
>
> ## On-policy error is not used (EoiX, ZE7F)
>
> ### Errors are absent in SFT
> We suggest that the distinction between “on-policy” error and “off-policy” errors is blurred because all errors are foreign to the training distribution because errors are absent from post-training. In practice, realism of errors may be more relevant because it has a higher probability of being generated. We provide 3 levels of realism in the Self-correction Bench, and they show meaningful correlation.
>
> ### On-policy error Validation
> [Additional experiment] To address this concern, we conducted experiments on on-policy error. We leverage the wrong completions (on-policy error) from ProcessBench[1], generated by Qwen2-7B-Instruct, Qwen2.5-7B-Instruct, Llama-3.1-8B-Instruct across GSM8K, MATH, OlympiadBench and OmniMath. To follow the same setting as in our framework, we pass the question and wrong generation to the user prompt, without any prompt engineering.
>
> We observe that models can correct these on-policy errors when presented as external errors (in the user prompt), achieving accuracies ranging from 14% to 59% across datasets.
> However, when these same errors originate from the model's own generation (internal errors), the blind spot would prevent this correction, resulting in lower accuracy. This validates that the blind spot phenomenon exists even for on-policy errors.
> | model                            | dataset       |   accuracy |   standard_error |   size |
> |:---------------------------------|:--------------|-----------:|-----------------:|-------:|
> | Qwen2-7B-Instruct           | gsm8k         |     0.4839 |           0.0912 |     31 |
> |           | math          |     0.2432 |           0.0502 |     74 |
> |           | olympiadbench |     0.4    |           0.0739 |     45 |
> |           | omnimath      |     0.1458 |           0.0515 |     48 |
> | Qwen2.5-7B-Instruct         | gsm8k         |     0.5833 |           0.1486 |     12 |
> |         | math          |     0.3462 |           0.0666 |     52 |
> |         | olympiadbench |     0.3036 |           0.062  |     56 |
> |         | omnimath      |     0.1731 |           0.053  |     52 |
> | Llama-3.1-8B-Instruct | gsm8k         |     0.5946 |           0.0818 |     37 |
> | | math          |     0.3382 |           0.0578 |     68 |
> | | olympiadbench |     0.2963 |           0.0511 |     81 |
> | | omnimath      |     0.1477 |           0.038  |     88 |
>
> The small sample size also echoes the data inefficiency of sampling on-policy error as model capability increases.
>
> ### Benefit of Off Policy Error
> When we designed our methodology, we have considered using on-policy error, but we found benefits of off-policy error outweighs its limitation:
> - Isolation of knowledge. By using off-policy, we could isolate confounding factors such as internal knowledge by design.
>
> - Cross model comparison. Like other LLM benchmarks, we could compare models side by side. Indeed, most benchmarks may not be “on-policy” for all models (e.g AIME for models not trained with AIME type questions).
>
> - Controlled distribution. We could control the targeted error, in our case, error in mathematical reasoning. It could be extended to other domains.
>
> - Practical challenge. As LLM is getting better, errors become rare. It is complicated by the fact that when a model is re-trained, the benchmark has to be re-generated.
>
> - Simulation of off-policy error in production. In production, foreign text, that might contain errors, would be fed to the model in tool use, for example.
>
> Yet we admit the limitation that while off-policy error can proxy on-policy error for practicality, its magnitude of self correction blind spot could be biased.
> Nevertheless, our benchmark introduces 3 levels of realism, progressing from artificial to realistic error. In Figure 3, we show that there exists a meaningful positive correlation of the blind spot as we progress from one level to a higher level of realism (e.g. SCLI->GSM8K_SC: 0.67, GSM8K_SC->PRM800K_SC: 0.62). As the realism of error increases, the off-policy error is closer to on-policy error, allowing reasonable extrapolation to on-policy scenarios.
>
> To conclude, both on-policy and off-policy errors are foreign to SFT data trained predominantly on error-free demonstrations, making the distinction less critical than error realism. We provide empirical evidence across 3 model families that the blind spot exists for on-policy errors. We also discuss the benefits of using off-policy error and the limitation.
>
> **Action**: We will include our additional experiments, the discussion of on-policy vs off-policy error and the limitation of using off-policy error in our manuscript.
>
> Again, we would like to thank the reviewers for the valuable feedback. We commit to harness them to enhance our manuscript. We hope that they have addressed these concerns.
>
> [1] Zheng. Processbench: Identifying process errors in mathematical reasoning, 2025. URL https://arxiv. org/abs/2412.06559.

---

> > ### Author Response · Authors · 2025-11-27
> > **Update to On-policy Error Validation Additional Experiment**
> >
> > Dear reviewers,
> >
> > **Correction**: We identified an implementation error in our previous response regarding on-policy error validation. The initial analysis used an incorrect prompt format - truncating reasoning traces at the first error rather than presenting full traces (minus answer) as in our main benchmark. We have re-run the analysis with the correct implementation.
> >
> > **Per-dataset results:**
> > | model                 | dataset       |   accuracy |   standard_error |   size |
> > |:----------------------|:--------------|-----------:|-----------------:|-------:|
> > | Qwen2-7B-Instruct     | GSM8K         |     0.0968 |           0.054  |     31 |
> > |      | MATH          |     0.1081 |           0.0363 |     74 |
> > |      | OlympiadBench |     0.0222 |           0.0222 |     45 |
> > |      | Omni-Math     |     0      |           0      |     48 |
> > | Qwen2.5-7B-Instruct   | GSM8K         |     0.1667 |           0.1124 |     12 |
> > |   | MATH          |     0.1154 |           0.0447 |     52 |
> > |    | OlympiadBench |     0.0357 |           0.025  |     56 |
> > |    | Omni-Math     |     0.0192 |           0.0192 |     52 |
> > | Llama-3.1-8B-Instruct | GSM8K         |     0.1622 |           0.0614 |     37 |
> > |  | MATH          |     0.1029 |           0.0371 |     68 |
> > |  | OlympiadBench |     0.0988 |           0.0334 |     81 |
> > |  | Omni-Math     |     0.0682 |           0.027  |     88 |
> >
> > **Aggregated results across datasets due to limited sample size:**
> >
> > | model                 |   size | Accuracy (95% CI)   |
> > |:----------------------|-------:|:--------------------|
> > | Qwen2-7B-Instruct     |    198 | 0.0606 ± 0.0333     |
> > | Qwen2.5-7B-Instruct   |    172 | 0.0640 ± 0.0367     |
> > | Llama-3.1-8B-Instruct |    274 | 0.0985 ± 0.0354     |
> >
> > **Interpretation**: While the correction rates (6-10%) are lower than initially reported, they demonstrate that models possess some capability to identify errors in their own reasoning when those errors are presented in user prompt. This is consistent with our core mechanism: blind spots stem from activation failure rather than pure knowledge deficiency.
> >
> > The low accuracies reflect two factors. First, ProcessBench samples particularly challenging problems (eg. OlympiadBench, Omni-MATH) where even external correction is difficult for 7-8B models. Second, on-policy errors necessarily conflate correction capability with knowledge limitations, which is the precise issue our controlled off-policy benchmark was designed to avoid.
> >
> > Importantly, this correction does not affect our manuscript and other experiments. This additional experiment was intended to demonstrate that blind spots exist for model-generated errors, which the corrected results continue to confirm through non-zero aggregated correction rates (6-10%).
> >
> > We appreciate the opportunity to provide corrected results and maintain transparency.

---

### Author Response · Authors · 2025-12-03
**Summary of Key Changes to Manuscript**

We sincerely thank the reviewers for their time and constructive feedbacks. We believe the manuscript has been significantly strengthened by addressing the reviewers' concerns.

In response to the feedbacks, we have conducted **4 additional experiments** and revised the manuscript to provide deeper causal analysis and broader validation. Below is a summary of the key changes and how they address the concerns raised:

# 1. Depth of Analysis and Contribution (Reviewer EoiX, 57Bc, DyNA)
**Concern**: Reviewers requested a deeper understanding of why this happens beyond "prompt engineering".

**Revision and Result**:
- (New Section 6.4): We moved beyond observation to obtaining causal evidence. We analyzed post training data to show the near-total absence of correction markers. Crucially, we evaluate the blind spot in DeepSeek-R1-Distill 8B and 70B models which are fine-tuned with error-correction traces generated by DeepSeek-R1. We note a reduction of the blind spot by 84.1% compared to base model, providing causal evidence that the blind spot is a fixable artifact of standard SFT data composition, offering a concrete training recipe (i.e. to include error and self-correction data in SFT) for the community.

# 2. On-Policy Errors (Reviewer EoiX, ZE7F)
**Concern**: Reviewers queried why on-policy error is not used and whether the blind spot phenomenon holds for errors generated by the model itself (on-policy) rather than injected errors.

**Revision and Result**:
- (Update to Section 8, previously Section 7): We discuss the methodology tradeoff when using off-policy error vs on-policy error. The primary merit is to isolate self-correction from knowledge limitation
- (New Section 7.3) We conducted a new experiment using ProcessBench[1] to test models on their own generated errors. We empirically confirmed that models can correct their own errors (statistically significant non-zero accuracy of 6-10% or 10-12% when excluding challenging maths problem) when presented as external user prompts but fail to do so when presented as their own reasoning. This confirms the Self-Correction Blind Spot exists in on-policy distribution.
- We redefined "own errors" as "internal errors" to avoid semantic debates and explicitly stated the off-policy nature of our error injection, addressing concerns about what constitutes "self-correction".

# 3. Generalization to Closed-Source Models (Reviewer 57Bc) & New Domains (Reviewer DyNA)
**Concern**: The lack of testing in closed-source frontier model and other domain beyond mathematical reasoning

**Revision and Result**:
- (New Sections 7.1) We evaluated Claude 3.5 Sonnet and Haiku, confirming the blind spot persists (52.5% and 41.4% respectively).
- (New Sections 7.2) We expanded to BIG-Bench Mistake[2] (Logical Deduction, Object Tracking), showing a blind spot of 62.8%, proving the phenomenon is domain-agnostic and not specific to mathematics.

# 4. Others:
- We have also updated the abstract and introduction reflecting the above changes.
- We included dataset examples (Appendix E), strengthened related work (e.g., citing N-Critics for "Wait" context) to address feedback

We believe these revisions transform the paper from an observational study into a robust analysis of a fundamental LLM behavior with re-usable benchmark and actionable solutions. We are grateful for the opportunity to improve our work and hope the revised manuscript has addressed most concerns.

# Reference
[1] Zheng et al. Processbench: Identifying process errors in mathematical reasoning, 2025. URL https://arxiv. org/abs/2412.06559.

[2] G Tyen et al. LLMs cannot find reasoning errors, but can correct them given the error location. Findings of the Association for Computational Linguistics: ACL.

---

### Meta-Review · Area_Chair_VFSP · 2026-01-10

**Summary:**

The paper introduces Self-Correction Bench, a controlled evaluation framework for measuring a self-correction blind spot, where LLMs correct errors more reliably when the same error is attributed externally (e.g., user/tool) than when it is framed as the model’s own output. The rebuttal adds additional experiments (including some closed-source and non-math settings, and on-policy validation) and argues a training-level mitigation via error-correction fine-tuning. Overall, while the benchmark/phenomenon is interesting and potentially useful, the core contribution remains largely empirical and prompt-driven, with unresolved concerns about off-policy construction, novelty, and limited technical depth, leading to a rejection recommendation given the initial negative scores (2,2,4,4).

**Reviewer Concerns:**

Initially, reviewers raised three main concerns. First, the benchmark relies heavily on off-policy injected errors (often generated by another model), weakening the claim of true “self”-correction; the rebuttal adds an on-policy validation, but after correcting an implementation mistake the evidence becomes weaker (still non-zero, but modest), so the core critique remains. Second, reviewers questioned depth/novelty, viewing the work as largely benchmark + prompt engineering; the general response strengthens a training-data mechanism and adds training-level evidence, but causality/mechanistic grounding is still not fully established. Third, reviewers flagged limited scope and evaluation, including initially math-focused coverage, missing closed-source results, and reliance on LLM-as-judge; the rebuttal broadens to Claude and some non-math tasks and clarifies checks, yet generality and benchmark validation remain only partially addressed.

**Reviewer Scores:**

Initially, the reviewers’ scores are (2,2,4,4). After the rebuttal, the two rejecting reviewers are unlikely to change their positions because their core concerns (off-policy validity + limited depth/novelty) remain central, and the corrected on-policy validation results are weaker than initially reported. The two borderline reviewers may credit the added results (closed-source + non-math) and the strengthened training-level argument, but the paper likely remains below threshold given the remaining fundamental issues. Overall, the expected post-rebuttal scores remain approximately (2,2,4,4) (at best (2,4,4,4)), and the submission still leans to rejection due to remaining core concerns.

---

### Decision · Program_Chairs · 2026-01-26

Reject